# Temporal evolution of master regulator Crp identifies pyrimidines as catabolite modulator factors

Ida Lauritsen [1,3], Pernille Ott Frendorf [1,3], Silvia Capucci[1], Sophia A. H. Heyde [1], Sarah D. Blomquist[1], Sofie Wendel[1], Emil C. Fischer[1], Agnieszka Sekowska[2], Antoine Danchin [2] & Morten H. H. Nørholm [1✉]

The evolution of microorganisms often involves changes of unclear relevance, such as transient phenotypes and sequential development of multiple adaptive mutations in hotspot genes. Previously, we showed that ageing colonies of an *E. coli* mutant unable to produce cAMP when grown on maltose, accumulated mutations in the *crp* gene (encoding a global transcription factor) and in genes involved in pyrimidine metabolism such as *cmk*; combined mutations in both *crp* and *cmk* enabled fermentation of maltose (which usually requires cAMP-mediated Crp activation for catabolic pathway expression). Here, we study the sequential generation of hotspot mutations in those genes, and uncover a regulatory role of pyrimidine nucleosides in carbon catabolism. Cytidine binds to the cytidine regulator CytR, modifies the expression of sigma factor 32 (RpoH), and thereby impacts global gene expression. In addition, cytidine binds and activates a Crp mutant directly, thus modulating catabolic pathway expression, and could be the catabolite modulating factor whose existence was suggested by Jacques Monod and colleagues in 1976. Therefore, transcription factor Crp appears to work in concert with CytR and RpoH, serving a dual role in sensing both carbon availability and metabolic flux towards DNA and RNA. Our findings show how certain alterations in metabolite concentrations (associated with colony ageing and/or due to mutations in metabolic or regulatory genes) can drive the evolution in non-growing cells.

[1] Novo Nordisk Foundation Center for Biosustainability, Technical University of Denmark, Lyngby, Denmark. [2] Kodikos Labs, Institut Cochin, Paris, France. [3] These authors contributed equally: Ida Lauritsen, Pernille Ott Frendorf. ✉email: morno@biosustain.dtu.dk

The mechanisms underlying the evolution of asexually reproducing microorganisms share many similarities with the development of cancer in multicellular organisms[1]. Clones isolated from cancer or microbial populations can be studied to explore the effect of individual driver mutations, but the true effect of these mutations may be difficult to delineate, when the experimental conditions inadequately mimic the environment they evolved from. Furthermore, much of our fundamental knowledge on evolutionary dynamics comes from studying well-mixed liquid microbial cultures, such as in Lenski's long-term evolution experiment[2], but these systems lack the 3D-structural component typical of e.g. the neoplasm in multicellular organisms, microbial colonies, or biofilm[3]. Thus, it is of general interest to study how mutations develop directly in structured environments such as ageing bacterial colonies.

Multiple sequential mutations are hallmarks of cancer development[1] and these often occur in hotspot genes. However, it is continuously debated to what extent hotspots represent mainly driver mutations or include a significant proportion of mutations that hitchhike along as passenger mutations[4]. Similarly, multiple mutations often appear in evolved microbial populations and this has stirred controversy because they can appear directed to specific genetic loci in apparent violation of the (neo-)Darwinian evolution theory[5–8]. We previously explored the evolution of ageing Escherichia coli (E. coli) colonies using a cya genetic background deficient in the synthesis of the signaling molecule cyclic AMP (cAMP), and observed mutations in several hotspot genes[9].

cAMP plays a major role in carbon catabolite repression (CCR) in E. coli: cAMP is produced in the absence of glucose[10], and then binds and activates the global transcription factor cAMP receptor protein (Crp, also known as Cap, Fig. 1a)[11,12] activating hundreds of genetic programs enabling e.g. growth on alternative carbon sources such as maltose or lactose. The cya mutant grows poorly but forms small white colonies on MacConkey agar supplied with maltose. When left in the incubator for up to two months, adaptive mutants appear as red papillae (secondary colonies, Fig. 1b, Supplementary Movie 1). Many of these have uncovered a cAMP-independent route to maltose fermentation, leading to the formation of organic acids and red colour due to a pH indicator in the medium[9].

A major mutational hotspot identified in this papillation assay is the crp locus and many of these mutants (termed Crp*) were shown to activate Crp in the absence of cAMP[13–15]. In a total of 594 clones previously sequenced, either by whole genome resequencing or by amplicon sequencing of the crp locus, 88% of the isolates had at least one mutation in crp and 15% had two mutations in crp[9] (Supplementary Table 1). Remarkably, one specific mutation was predominant: an alanine to threonine substitution in position 144, occurring in 67% of the isolates with mutations in crp. The second most abundant mutation identified was in the same position and changed alanine into glutamate (17% of the isolates with crp mutations). A144T is a canonical Crp* mutant and together with G141D, it is the most frequently isolated Crp mutation – both have been identified in at least four independent in vivo evolution experiments since 1981 (reviewed in[15]), and the structure of the A144T mutant has been studied using X-ray crystallography[16]. A144 is located in the D alpha-helix of Crp's carboxy domain, near the hinge connecting the cAMP and the DNA binding domains, and the A144T mutation renders Crp partially cAMP-independent and activatable by other nucleotides such as cGMP, adenosine, and AMP[14,17].

Another mutational hotspot, identified in the 96 full genomes sequenced from papillae, was the cmk locus[9]. cmk encodes cytidylate kinase that catalyses the phosphorylation of the pyrimidine nucleotide CMP to CDP[18] (Fig. 1c). cmk mutations were found in 24 of the 96 sequenced genomes and always co-occurred with the crp mutations A144T, A144E, or T140R[9]. Mutations were also identified in carA, udk, pyrC, pyrG, pnp, and umpH[9], strongly indicating a phenotypic link between pyrimidine metabolism and Crp-dependent growth on maltose. Living cells must maintain nucleotide homeostasis in response to the supply and demand of genetic material. This is critical because the concentration and balance of nucleotides affect mutation rates[19]. All purine nucleotides are synthesized de novo, whereas only the pyrimidine UMP is synthesized directly from central metabolism[20]. The dephosphorylated nucleoside uridine and the nucleobase uracil are not precursors of UMP in this biosynthetic pathway (Supplementary Fig. 1). However, enzymes exist that enable scavenging of uridine and uracil from the environment or from the turnover of RNA that is converted into UMP. In E. coli, the two enzymes UmpH and UmpG produce uridine from UMP as a safety mechanism termed directed overflow metabolism to maintain pyrimidine homeostasis when genetic material is in low demand[21]. Cytidine, CMP, and CDP are even further disconnected from de novo synthesis and are produced in all living cells only by hydrolysis of RNA, of precursors of complex carbohydrates or phospholipids, or scavenging from the environment[22] (Supplementary Fig. 1), noting that a cytosine scavenging enzyme such as cytosine phosphoribosyltransferase has not yet been identified in any extant organism[23]. CDP is the precursor of deoxy CDP that is required for incorporating cytosine into DNA and is a major source of thymine (Supplementary Fig. 1). This raises fundamental questions: why is pyrimidine biosynthesis conserved like this, and do pyrimidines serve special roles in living cells?

Here, we show that the sequential generation of multiple driver mutations in crp is affected both by pyrimidine metabolites that naturally build up in the ageing bacterial colonies and by the hotspot mutations that occur in genes involved in pyrimidine metabolism.

## Results

**The dominating Crp* mutation displays a transient phenotype**. While a number of the strains isolated in the cya papillation experiment exhibited a clear maltose fermentation phenotype, curiously, the dominating CrpA144T mutation exhibited only a transient phenotype on the selective medium. Mutant papillae typically turn red on MacConkey agar when maltose is efficiently fermented (Fig. 1b), and the A144T mutant grew better than the parental strain (Fig. 1d). However, the red colour phenotype was gradually lost when A144T mutants were restreaked on fresh medium (Fig. 1e).

**Additional mutations develop sequentially in crp**. In the 594 sequenced crp loci, some mutations such as Q170K and S62F occur at above-average frequency but are only found in combination with other mutations – mainly A144T, but also A144E, and T140R[9] (Supplementary Table 1). In line with this observation, papillae occurred at high frequency when starting with a crpA144T strain background (Fig. 1f) and deep sequencing of papillae developing from this background confirmed the appearance of mutations Q170K, S62F, and many others[15]. Altogether, these observations suggest that the additional crp mutations are not merely passenger mutations, hitchhiking along with Crp* mutations. Other canonical Crp* mutations such as T140K and G141D develop as single Crp mutations, but at a much lower frequency than A144T[9]. What makes A144T dominant over other Crp* mutations under these selective conditions, and why do second site mutations develop sequentially in the A144T background?

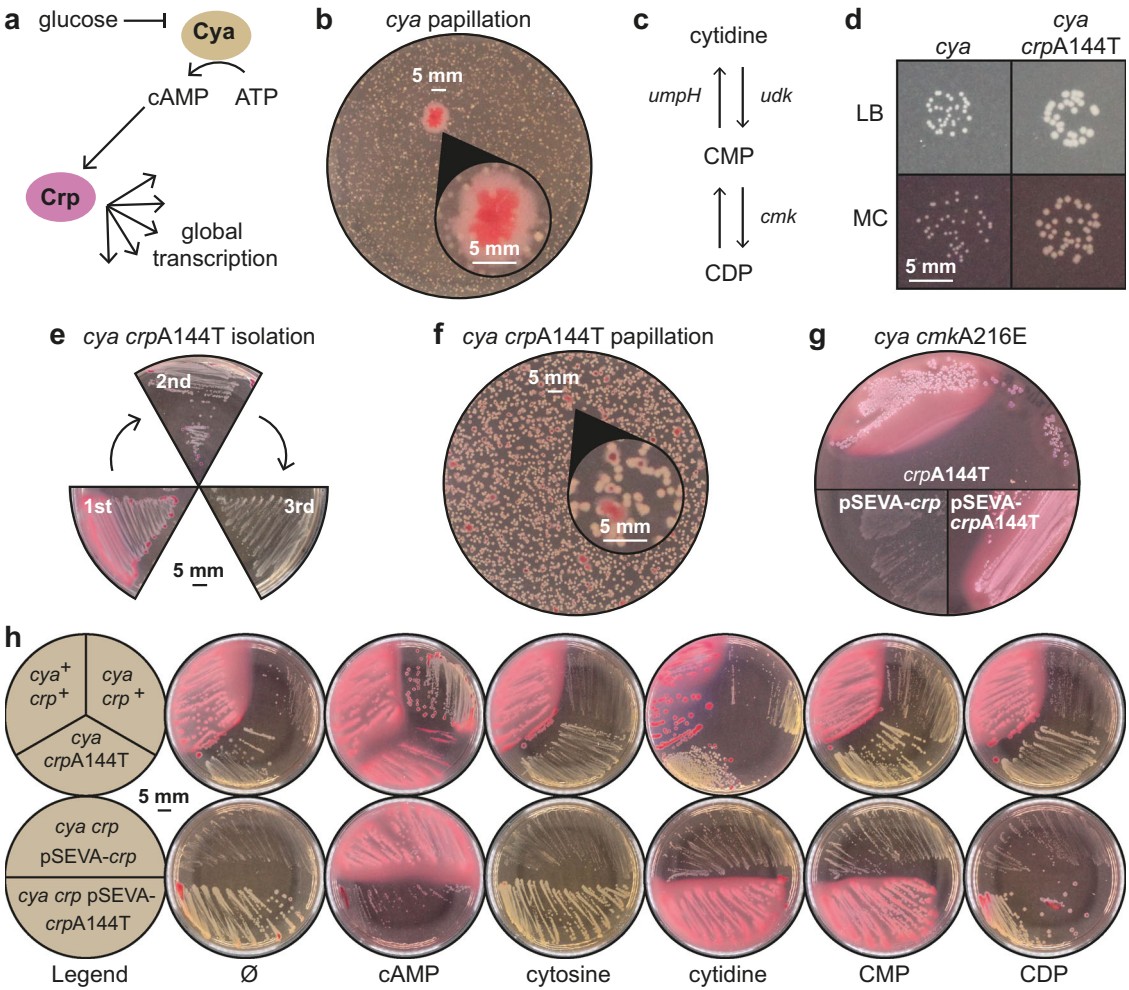

**Fig. 1 Crp mutants point to a link between carbon catabolite repression and pyrimidine metabolism. a** In the absence of glucose, the enzyme adenylate cyclase (Cya) produces cyclic adenosine monophosphate (cAMP) that binds and activates the global transcription factor cAMP Receptor Protein (Crp). **b** *Escherichia coli cya* mutants grow poorly on maltose MacConkey medium, but upon extended incubation mutant red (fermenting) papillae appear. **c** Synthesis and turnover of the pyrimidine nucleoside cytidine in *E. coli*. The cmk gene encodes cytidylate kinase that converts cytidine monophosphate (CMP) into cytidine diphosphate (CDP). CMP can also be converted into cytidine by the action of the ribonucleotide monophosphatase UmpH. Cytidine is phophorylated by the kinase Udk. **d** A majority of papillae obtain the mutation CrpA144T and the mutant exhibits a moderate increase in growth on both LB and maltose MacConkey agar plates (MC). **e** Mutants such as CrpA144T typically appear as red papillae because fermentation acidifies the medium, but often the fermentation phenotype gradually disappears upon re-streaking clones. **f** The *crp*A144T mutation accelerates mutant papillae formation that arise from almost every colony after few days. **g** A CmkA216E mutation frequently identified in papillae increases the activity of CrpA144T. This is shown in a *cya crp*A144T strain background (upper part of the plate) or in a *cya crp* background that is complemented with low-copy plasmids expressing either wildtype *crp* or the A144T mutant (lower part of plate). **h** Exogenously added cytosine nucleosides and nucleotides stimulate the activity of the CrpA144T mutant, both when expressed from the genome (pink single colonies with cytidine) and from a low-copy pSEVA plasmid (significant acidification both with cytidine and CMP). cAMP was included as a positive control and no supplement (Ø) served as a negative control.

**The dominating *cmk* mutation improves maltose fermentation in combination with a *crp*A144T mutant**. We previously showed that when attempting to introduce the dominating *cmk* mutation (A216E) into the parental *cya* strain by recombineering, we were unable to isolate strains without A144T mutations spontaneously forming in *crp*, and that the cya *crp*A144T *cmk*A216E triple mutant fermented maltose more efficiently than the cya *crp*A144T double mutant[9]. To further study the possible link between *crp* and pyrimidine metabolism, we deleted *crp* from a *cya cmk*A216E strain, reintroduced *crp* and *crp*A144T on low-copy plasmids, and monitored growth on maltose MacConkey agar. This confirmed that CrpA144T appeared more active in a *cmk*A216E background compared to a *cmk* + background, both when *crp*A144T is expressed from the genomic locus and from a

plasmid (Fig. 1g, h). What is the connection between Crp and hotspot mutations related to pyrimidine metabolism?

**Cytidine and CMP promote maltose-dependent growth**. We hypothesized that a metabolite in the spatiotemporal micro-environment, perhaps related to pyrimidines, was building up in the ageing colonies, and was affecting the evolutionary solution space of Crp mutants isolated. The rationale was that upon restoring growth, either by re-streaking on fresh medium or by forming a Crp* mutation in a subpopulation of the colony, the metabolite could be gradually lost together with the maltose fermentation phenotype. At the same time, additional mutations conferring independence of this metabolite would become advantageous.

Several of the *cmk* mutations introduced stop codons in the *cmk* gene[9] and *cmk* knockout mutants have been shown to accumulate 30-fold more CMP than a wildtype strain[18]. We therefore speculated that CMP, or a closely related metabolite, was playing a role in the promotion of cAMP-independent growth. To test this hypothesis, we plated a *cya crp*A144T double mutant, a *cya* single mutant, and a *cya* + strain on maltose MacConkey agar in the presence of cytosine, cytidine, CMP, CDP, or cAMP and monitored growth on maltose MacConkey agar. We observed strong acidification of the medium by the *cya crp*A144T mutant only when grown in the presence of cAMP, but also weak acidification when grown with cytidine (Fig. 1h). To substantiate this observation, and because additional mutations occur at higher frequency in this background making it hard to control genetically (Fig. 1f), we complemented a *cya crp* strain with the low-copy plasmid version of wildtype *crp* or *crp*A144T and observed a clear fermentation phenotype of the A144T mutant when grown in the presence of cytidine or CMP (Fig. 1h). The different phenotypes of the *crp*A144T mutants, observed when expressed from the genome or the low-copy vector, is likely due to higher expression of *crp* from the plasmid, which was confirmed on the protein level by western blotting and on the RNA level by RNA sequencing (Supplementary Fig. 2). This is probably also why *crp*A144T expressed from a plasmid becomes toxic in presence of cAMP (Fig. 1h).

**Nucleotides accumulate in ageing bacteria**. To explore the physiological relevance and potential change in nucleotide levels under the experimental conditions explored here and in the different mutants isolated, we plated *cya*, *cya crp*, *cya crp cmkA216E*, and *cya crp cmk* mutants on maltose MacConkey agar, extracted metabolites from the cells from one-day and five-day-old cultures, and analysed the extracts by LC-MS (Fig. 2a). Whereas the large majority of metabolites identified varied very little in concentration between sampling days and genetic backgrounds (Supplementary Data 1), CMP increased two-to-three fold from day one to day five in the *cya* and *cya crp* strains and was significantly higher in the mutant *cmk* strains (Fig. 2b). This confirms previous observations that CMP levels increase in *cmk* knockouts and indicates that the dominating *cmk*A216E mutation is causing reduced Cmk activity.

With this approach, we observed highly variable levels of the cytidine nucleoside and speculated that this could be due to secretion. To analyse the sum of nucleosides inside the cells and secreted into the medium, we repeated the experiment in liquid culture in a medium highly similar in composition to MacConkey agar, and analysed cell extracts together with the medium. This time we decided to analyse the cytidine content in wildtype, *cya*, and *cya crp* cells after one, three, and six days of incubation. Cytidine levels varied less across replicates with this approach and showed a drastic 26-33 fold increase in the *cya* and *cya crp* mutants (Fig. 2c) from day one to day six. Cytidine also increased in the wildtype strain, but only five-fold at day six compared to day one. Interestingly, this metabolomics analysis of ageing bacteria showed that, compared to other metabolites, nucleosides/nucleotides in general had the highest concentration increase over the time course of the experiment (Fig. 2d) and cAMP was the metabolite that increased the most in the wildtype strain (Supplementary Fig. 3), highlighting the well-known significance of this signaling molecule in bacterial physiology. Combined, these experiments revealed a marked increase in cytosine nucleoside/nucleotide levels five-to-six days after incubation in MacConkey-like media, in particular in *cya* and *cmk* mutants, supporting that these metabolites could play a role in the selection of *crp* mutants. It is interesting to note that five days is exactly the time when papillae started appearing in the original experiment[9].

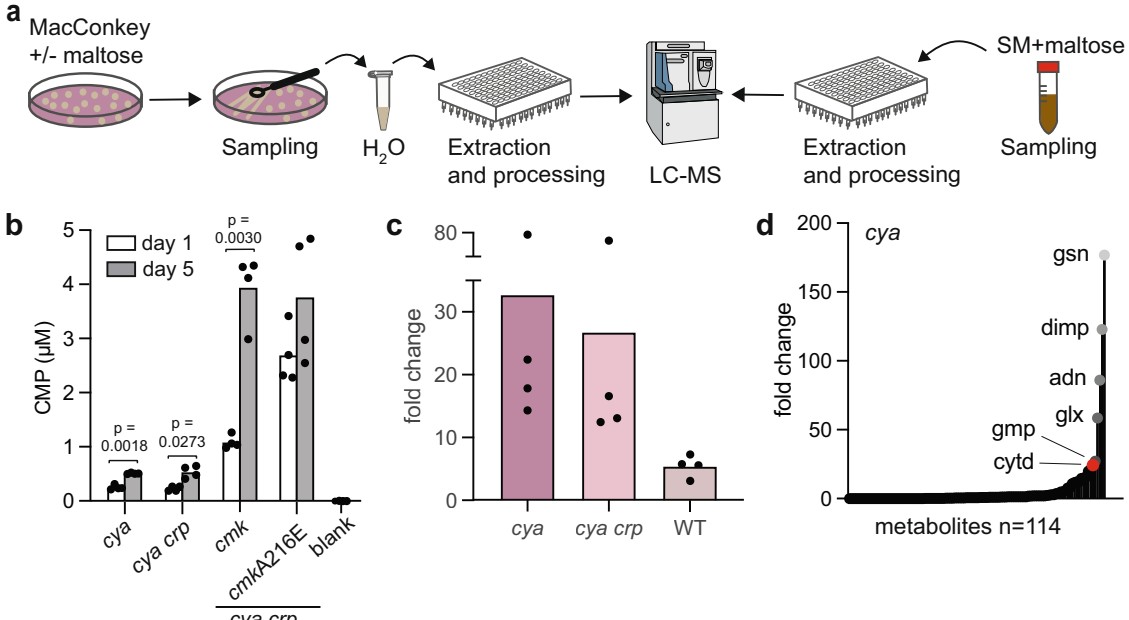

**Fig. 2 Cytidine and CMP accumulate in ageing bacteria. a** Schematic overview of biomass sampling from MacConkey agar plates or Simple MacConkey (SM) liquid cultures for metabolomics with liquid chromatography-mass spectrometry (LC-MS). **b** Absolute cytidine monophosphate (CMP) levels from sampling day one (white) and five (grey). Significance was based on two-sided paired t-tests between the groups indicated, and designated as significant if *p* < 0.05. Data represent four replicates. **c** Fold change of absolute cytidine levels in *cya*, *cya crp*, and WT (differentiated by color) for days one to six. Data represent four replicates. **d** Distribution of fold changes of all metabolites detected by LC-MS from day one to six for *cya*. The metabolites with the highest fold change are highlighted. From light to darker grey: gsn: guanosine, dimp: deoxyinosine monophosphate, adn: adenosine, glx: glyoxylate, gmp: guanosine monophosphate, cytd: cytidine (red). Data represents the average of four replicates. Source data are provided as Supplementary Data 1.

**Formation of mutant papillae is accelerated with increased levels of cytosine nucleosides / nucleotides.** Our inability to generate *cmk* mutants without *crp*A144T spontaneously forming after mutation of the genome indicates a strong selective pressure on *crp* when cytidine nucleotide levels are increased. To be able to better follow the evolution of *crp* under these conditions, we reintroduced *crp* on the low-copy plasmid into the *cya crp*, *cya crp cmk*, and *cya crp cmk*A216E strains and monitored growth and evolution on maltose MacConkey agar. After seven days in the incubator, no papillae had formed with empty vector control, whereas a few papillae formed in the presence of the *crp* plasmid (Fig. 3a). In contrast, pink secondary colonies formed from a large number of colonies in the mutant *cmk* strains with the *crp* plasmid. Similarly, the *cya crp* strain harboring the pSEVA-*crp* plasmid formed more papillae in the presence of exogenously added cytidine (Fig. 3b) and in both experiments the majority of these papillae were subsequently found to have developed the *crp*A144T mutation (Supplementary Table 2).

**Cytidine affects catabolism of other carbon sources.** Our findings strongly suggest a link between cytidine, Crp, and maltose utilization in the *cya* mutant, but does cytidine affect carbon catabolism more broadly and in a wildtype background? To explore this, we plated the wildtype *E. coli* K12 MG1655 on MacConkey medium supplemented with 11 different carbon sources in the absence or presence of cytidine. Acidification by fermentation of glycerol, sorbitol, galactose, and melibiose was severely inhibited by cytidine, small negative cytidine-effects were observed on maltose, ribose, rhamnose, and fucose, whereas no significant effects were observed with glucose, lactose, and xylose (Fig. 3c, d, Supplementary Fig. 4). This suggests that cytidine plays a more general role in carbon catabolism.

**Cytidine and uridine activate CrpA144T in vitro.** A drawback of these in vivo screens is that it is not possible to rule out that different nucleosides and their phosphorylated counterparts are transported differently across the inner and outer membrane of *E. coli*, or that phosphorylation or dephosphorylation of the added compounds take place. For example, the positive effect of CMP may be due to extracellular dephosphorylation and subsequent uptake of cytidine. To directly test interactions between Crp and different metabolites, we performed in vitro label-free biolayer interferometry (BLI). Wildtype Crp and A144T were expressed and purified by affinity chromatography, mixed with different ligands, and interactions with biotinylated synthetic DNA encoding P*malT*, a Crp responsive promoter from *E. coli*[24], were analysed using the Octet RED96 system (Fig. 4a). In line with our in vivo screen, this BLI analysis showed that cytidine and cAMP activated the DNA binding activity of the A144T mutant (Fig. 4a, b), as did uridine, but not the 2'-deoxy nucleoside thymidine, CMP, or UMP (Fig. 4b). In contrast, only cAMP activated wildtype Crp (Fig. 4c). This suggests that CrpA144T interacts directly with pyrimidine nucleosides.

**Cytidine and uridine activate CrpA144T, but inhibit wildtype Crp in vivo.** Cytidine and uridine have not previously been reported to interact with Crp. To explore this in more detail in vivo, we developed a Crp activity reporter with a high dynamic range based on a plasmid carrying a fusion between P*malT* and the fluorescent reporter GFP (Fig. 5a). As intended, the construct responded highly sensitively to different concentrations of cAMP added exogenously in a *cya* background (Fig. 5a). In agreement with both our in vitro data, and our qualitative in vivo assays on maltose MacConkey, we observed an increase in activity of the CrpA144T mutant in the presence of cytidine (Fig. 5b) and uridine (Supplementary Fig. 5). Also in line with the mild inhibitory effect on maltose MacConkey, in the wildtype K12

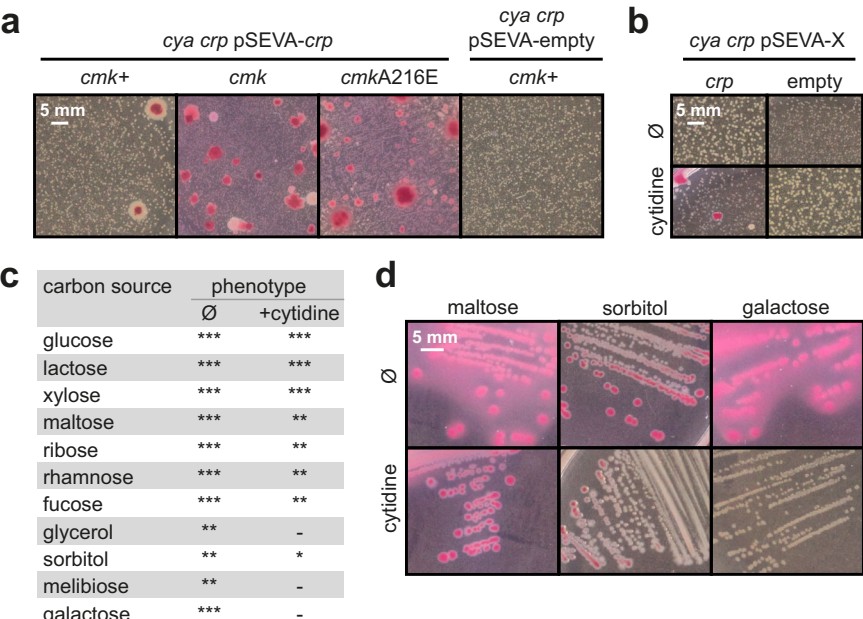

**Fig. 3 Influence of cytidine and *cmk* mutations on the phenotypes of *E. coli* on MacConkey agar supplied with different carbon sources. a** Papillation assay of *cya crp cmk* + , *cya crp cmk*, and *cya crp cmk*A216E strains on maltose with *crp* supplied on a low-copy plasmid or the empty plasmid as control. Pictures were taken after 7 days of incubation. **b** Papillation assay of the *cya crp* strain on maltose with or without *crp* supplied on a low-copy plasmid and with or without 10 mM cytidine. Pictures were taken after 3 days of incubation. **c** Phenotypes of the *E. coli* wildtype strain when supplemented with different carbon sources with or without 10 mM cytidine. Fermentation phenotypes are indicated by *** (red color and media acidification), ** (red color but no media acidification), * (limited red color), or - (white color). **d** Representative pictures of strains, where cytidine has an inhibiting effect on fermentation. Referenced phenotypes on all carbon sources can be found in Supplementary Fig. 4. Ø; no supplementation.

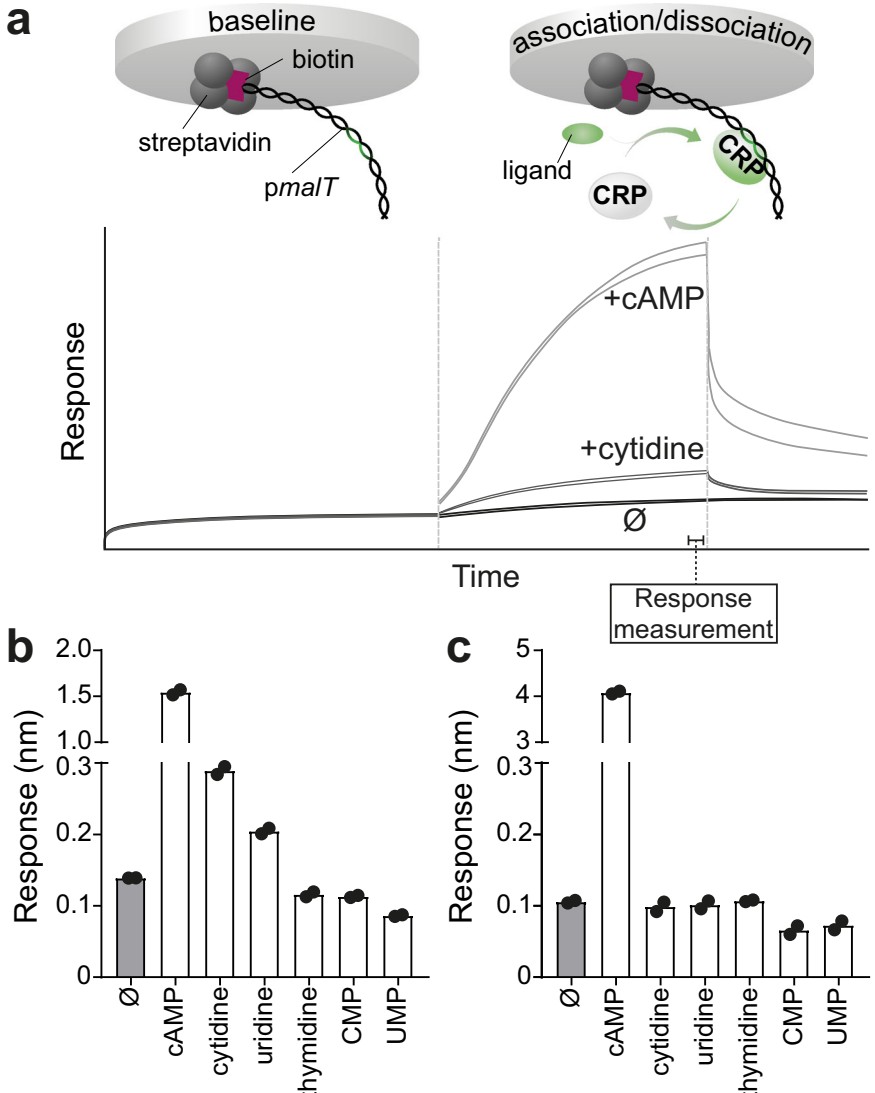

**Fig. 4 In vitro activity of Crp and CrpA144T using biolayer interferometry. a** Upper panel: Illustration of biolayer interferometry where a biosensor coated with streptavidin binds to biotinylated DNA encoding the Crp-responsive P*malT* promoter, and produces a signal when associating and dissociating with the Crp protein depending on the ligand present. Lower panel: Representative output showing the progression of output (nm) aligned to baseline. Data correspond to part of the data in panel 2b, in the absence of ligand (∅), with the positive control cyclic adenosine monophosphate (cAMP) or with cytidine. **b** Association of CrpA144T to P*malT* in the presence of pyrimidines (10 mM) or the positive control cAMP (0.5 mM). CMP, cytidine monophosphate; UMP, uridine monophosphate. Absence of ligand (∅) serves as a negative control. **c** Association of wildtype Crp to P*malT* in the presence of different pyrimidines, the positive control cAMP (white) or no ligand (grey, ∅). Data represent the average of two replicates.

MG1655 strain, the reporter was clearly inhibited by exogenously adding these pyrimidines, and we observed the same effects of cytidine supplementation when replacing P*malT* with P*lac* in the reporter plasmid (Fig. 5b, Supplementary Fig. 5). Although a fixed concentration of cytidine was applied in these experiments, we also found that the cytidine-response varied with the concentration (Supplementary Fig. 6) and we verified using transcriptomics that the observed effects of cytidine supplementation are not due to varying Crp levels (Supplementary Fig. 7). This suggests that cytidine and uridine act as signaling molecules by interfering with Crp. The observed inhibitory effect of cytidine appears stronger in vivo than in vitro, which may reflect missing components in the simplified in vitro system or that pyrimidines affect more than the binding of Crp to DNA, for example Crp-RNA polymerase interactions.

Uridine acts similarly to cytidine by inhibiting Crp and activating CrpA144T. However, in contrast to CMP, which is not,

UMP can be synthesized de novo (Supplementary Fig. 1) and biosynthesis is feedback regulated, which may explain why *cmk* mutations that likely lead to a build-up of both CMP and cytidine are more frequently observed under our experimental conditions. Accumulation of uracil and uridine, catalysed by the enzyme UmpH (Fig. 1c, Supplementary Fig. 1), was previously observed in mutant *E. coli* with defective feedback regulation of pyrimidine metabolism[21]. In nutrient-rich conditions uridine may be the more relevant Crp ligand as it is present at mM concentration levels[25]. To explore if Crp activity is affected by manipulating endogenous pyrimidine levels, we overexpressed?-P *umpH* from a plasmid in the presence of the P*malT*-GFP reporter. Indeed, *umpH* expression significantly reduced expression from the Crp sensitive reporter (Fig. 5c). In a previous study, global gene regulation was explored when an overflow of uridine and uracil occurred in a pyrimidine feedback dysregulated strain[21]. In line with our observations, transcriptomics showed that expression of

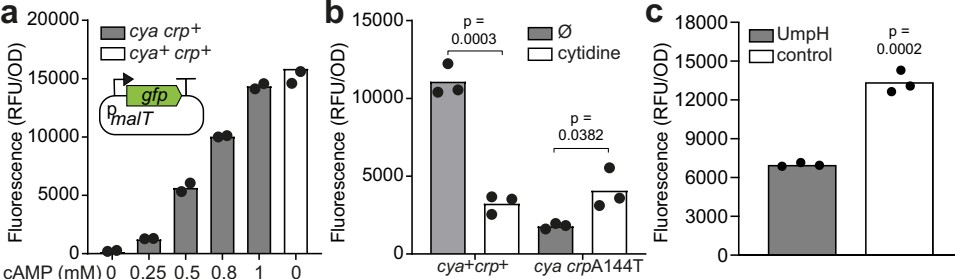

**Fig. 5 In vivo pyrimidine effects on Crp and CrpA144T determined with a sensitive Crp reporter. a** Illustration and validation of an in vivo Crp activity reporter based on a plasmid with the P*malT* promoter controlling expression of *gfp*. The reporter was transformed into a *cya* strain, and fluorescence was detected in the presence of different concentrations of cyclic adenosine monophosphate (cAMP). A *cya+* strain served as a positive control (white). Data represent the average of two biological replicates. **b** The effect of exogenously added cytidine (white) on the Crp activity reporter in a wildtype (*cya+crp+*) or mutant (*cya crp*A144T) strain background. No supplementation (grey, Ø) serves as a negative control. Data represent the average of three biological replicates. **c** Effect of overexpression?- P of *umpH* (grey), compared to empty vector control (white), in wildtype *E. coli* K12 MG1655 assayed with the GFP reporter. Data represent the average of three biological replicates. Significance was based on two-sided unpaired t-tests between the groups indicated, and designated as significant if *p* < 0.05. RFU, relative fluorescence units; OD, optical density.

genes known to be positively regulated by Crp (*yfcT*, *malE*, *malK*, *malM*, *bglB*, and *csgF*) were repressed in this strain, whereas genes known to be downregulated by Crp (*gadA*, *gadE*, *gadX*, and *gadW*) were upregulated[21]. Together, these observations suggest that endogenous pyrimidine levels affect Crp signaling.

**The mutation Q170K confers independence from cytidine.** Additional mutations such as Q170K develop sequentially in the *crp* hotspot and we speculated that this was due to the transient nature of e.g. the A144T mutant phenotype – i.e. the selective advantage of the A144T mutation might decrease if pyrimidine levels drop upon resuming growth from the ageing bacterial colony. To investigate this hypothesis, we assayed the activity of both the A144T mutant and the A144T Q170K double mutant expressed in the presence or absence of 10 mM cytidine using both the GFP reporter and maltose MacConkey agar. This showed that Q170K has an activating effect on CrpA144T, much like cytidine, and that the double mutant is no longer affected by cytidine (Fig. 6a, b). It is thus plausible that A144T is strongly selected for when pyrimidines accumulate, and that when growth is restored, pyrimidine levels drop, thereby making additional mutations such as Q170K advantageous.

**Canonical Crp\* mutations that occur at low frequency are not activated by cytidine.** To fully understand the temporal evolution of Crp, it is important not only to consider the mutations that dominate, but also those that are underrepresented: The A144T (GCA- > ACA) and A144E (GCA- > GAA) mutations occur at a higher frequency than other mutations such as T140K (ACG->AAG) and G141D (GGC->GAC)[9], but they have all been characterized previously as cAMP-independent Crp\* mutants[15]. Further, the mutated residues are neighbours in the same structural domain, and even pairwise represent the same types of mutations (C->A or G->A). The latter is important because the mutations C->A, most likely caused by oxidation of guanosine on the complementary strand, and G->A, most likely caused by cytosine deamination on the complementary strand, are highly dominant under these conditions[9,26] and mutation bias caused by the available mutational space could limit the observed solution space. This prompted us to assay the activity of these variants in response to different pyrimidine levels. We found that the T140K and G141D mutants were less active than A144T and were not activated by cytidine (Fig. 6c, d). Furthermore, a fitness assay showed a competitive advantage of A144T relative to T140K and

G141D in the presence of cytidine when grown on agar plates with maltose (Fig. 6e, f, Supplementary Fig. 8).

**The cytidine regulator CytR plays a role in carbon catabolism.** Apart from the apparent direct effect of cytidine (and uridine) on Crp, could other mechanisms explain the link between carbon and pyrimidine metabolism? Cytidine is known to bind to the cytidine regulator CytR and interact with Crp in a small regulon mostly involved in nucleotide metabolism[27]. To explore a potential involvement of CytR, we generated a *cya crp cytR* deletion strain and monitored growth on MacConkey agar supplied with either maltose or galactose in the presence of pSEVA-*crp*, pSEVA-*crp*A144T, or empty vector control. Compared to the *cytR +* strain, maltose fermentation was less stimulated by cytidine in the *cytR* strain when expressing *crp*A144T (Fig. 7). However, cytidine still negatively affected galactose fermentation in the *cytR* strain, both with *crp*A144T and when wildtype *crp* was activated by exogenously added cAMP. This suggests both CytR-dependent and CytR-independent effects of cytidine in carbon catabolism. Interestingly, the *cytR* mutant grew poorly on maltose MacConkey both in the absence and presence of wildtype *crp*, even when cAMP was supplied exogenously. In contrast, severe growth defects were only observed on galactose MacConkey when both *crp* and *cytR* was deleted, or with the toxic combination of CrpA144T and cAMP. This suggests a complex interplay between Crp, CytR, and nucleotide levels in carbon catabolism.

**Expression of the heat shock sigma factor RpoH is perturbed in the *cya* mutant.** Beyond the role in regulating genes involved in nucleotide metabolism, a clue to a more global regulatory role of CytR comes from its regulation of the heat shock sigma factor encoding *rpoH*[28]. To explore the potential role of *rpoH*, we constructed a P*rpoH* promoter *gfp* fusion and monitored the fluorescence from this reporter in the *cya crp* and the *cya crp cytR* strain in combination with the pSEVA-*crp* or pSEVA-*crp*A144T plasmids, and in the presence and absence of exogenously added cytidine and cAMP. Fluorescence from this reporter was completely absent in the *cya* strain with the wildtype *crp* plasmid, but was high both in the absence of *cytR* and with the combination of the CrpA144T mutant with cytidine in the *cytR +* background (Fig. 8a). This shows that P*rpoH* promoter activity is very low in the *cya* genetic background, confirms the previously observed regulation of *rpoH* by CytR and cytidine[28], and shows a large impact of the CrpA144T mutant on P*rpoH*.

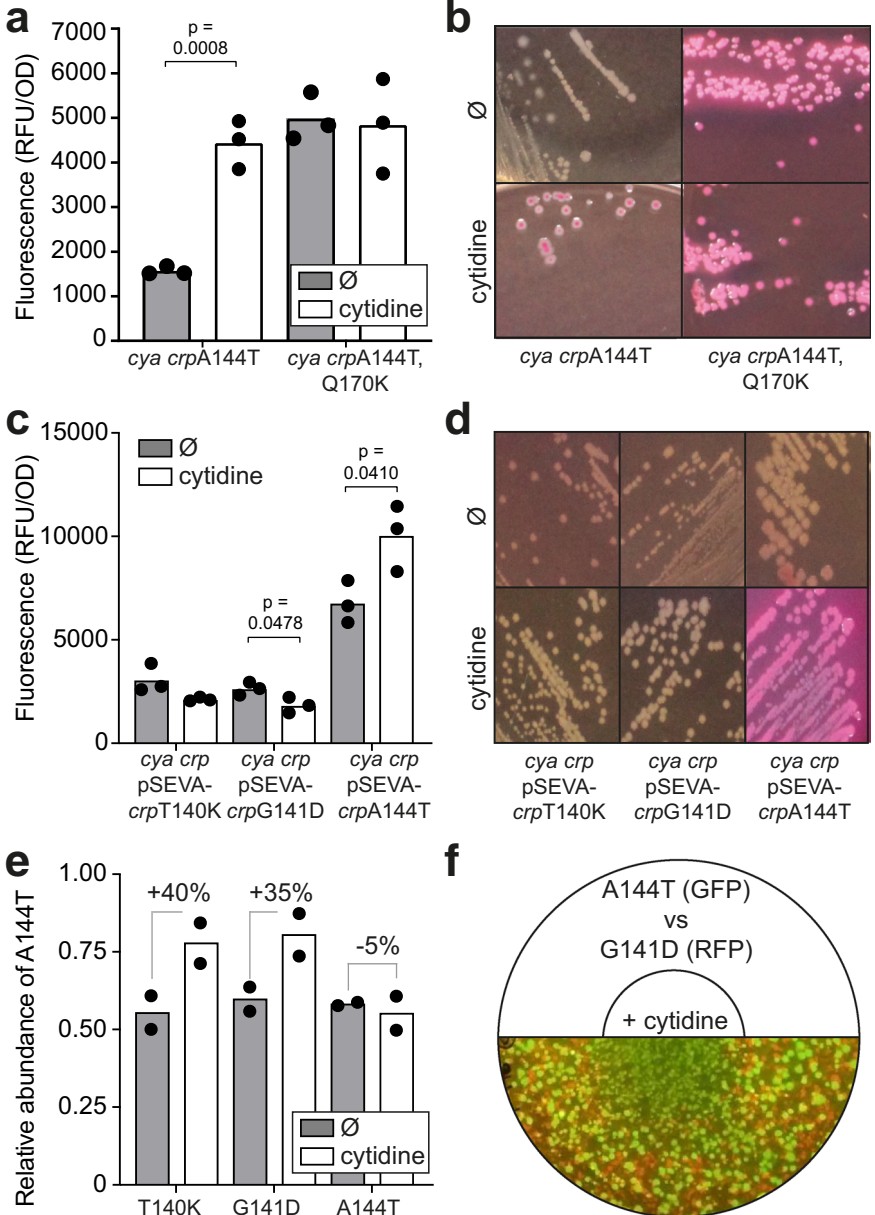

**Fig. 6 In vivo activity and fitness effects of different Crp mutants. a** Effect of the additional Crp mutation Q170K in the presence (white) or absence (grey) of exogenously added cytidine assayed with the GFP reporter. No supplementation (Ø) serves as a negative control. Data represent the average of three biological replicates. **b** Phenotypes of Crp A144T Q170K compared to A144T on maltose MacConkey agar. **c** Relative activities of Crp* mutations T140K and G141D compared to A144T, expressed from a low-copy SEVA plasmid, in the presence (white) or absence (grey) of cytidine assayed with the GFP reporter. Data represent the average of three biological replicates. **d** Phenotypes of the Crp* mutations T140K, G141D, and A144T on maltose MacConkey agar. **e** Relative abundance of strains carrying the Crp* mutations T140K and G141D in competition with A144T when grown on agar plates with maltose supplied with cytidine (white) or water (grey, Ø). RFP fluorescence measurements of cells taken from different sections of the agar plate were used to determine the cytidine diffusion gradient. The relative abundance was calculated based on the fluorescence levels for one strain expressing *rfp*. Data represent the average of two biological replicates. **f** Illustration and representative picture of competition assay on agar plates. The two competing strains, expressing either *gfp* or *rfp*, were mixed in equal proportions and a 10 µl drop of 0.5 mM cytidine or water was applied to the center of the agar plate for diffusion. Significance was based on two-sided unpaired t-tests between the groups indicated, and designated as significant if *p* < 0.05. RFU, relative fluorescence units; OD, optical density.

To further study the role of *rpoH*, we attempted to delete the gene in the different genetic backgrounds studied here. RpoH has previously been reported to be essential for growth at 37 °C, but not at temperatures below 30 °C[29]. Curiously, we were unable to isolate *rpoH* mutants in the wildtype K12 MG1655 background, but were successful in different *cya* mutants. This again gives clues to a tight interplay between Crp and RpoH in bacterial physiology, but further phenotypic characterizations of the *cya* *rpoH* mutant strains were inconclusive due to the poor fitness in general of the *rpoH* strains.

**A naturally occurring *cis*-acting mutation in the *malT* promoter makes it insensitive to cytidine.** Due to the complex *trans* effects when studying perturbations to global gene regulators such as Crp and RpoH, we finally returned to the original papillae evolution experiment to look for further clues to the regulation of

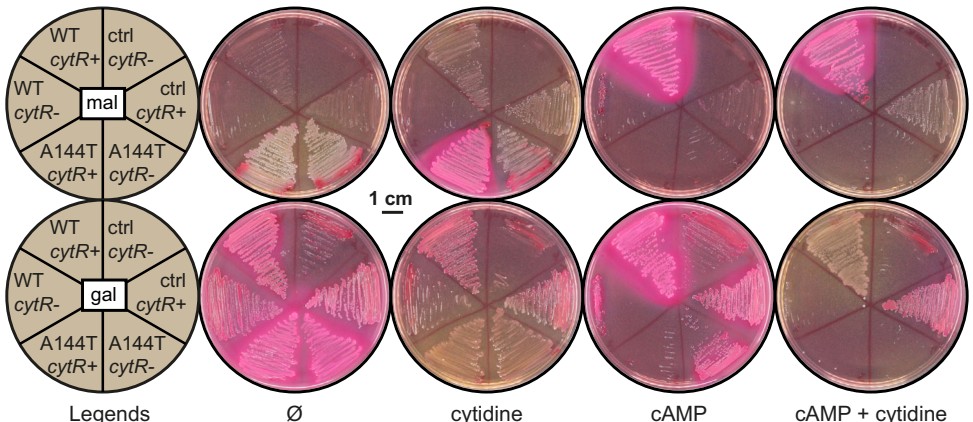

**Fig. 7 Phenotypes of *cytR* strains on MacConkey agar.** Fermentation phenotypes on MacConkey agar of MG1655 *cya crp cytR* (+/−) carrying the low-copy pSEVA plasmid expressing either wildtype *crp* (WT), the A144T mutant, or empty vector control (ctrl). The plates were supplied with maltose (mal) or galactose (gal) as a carbon source and supplemented with cytidine, cAMP or both, or without supplement (∅). Experiments were repeated three times.

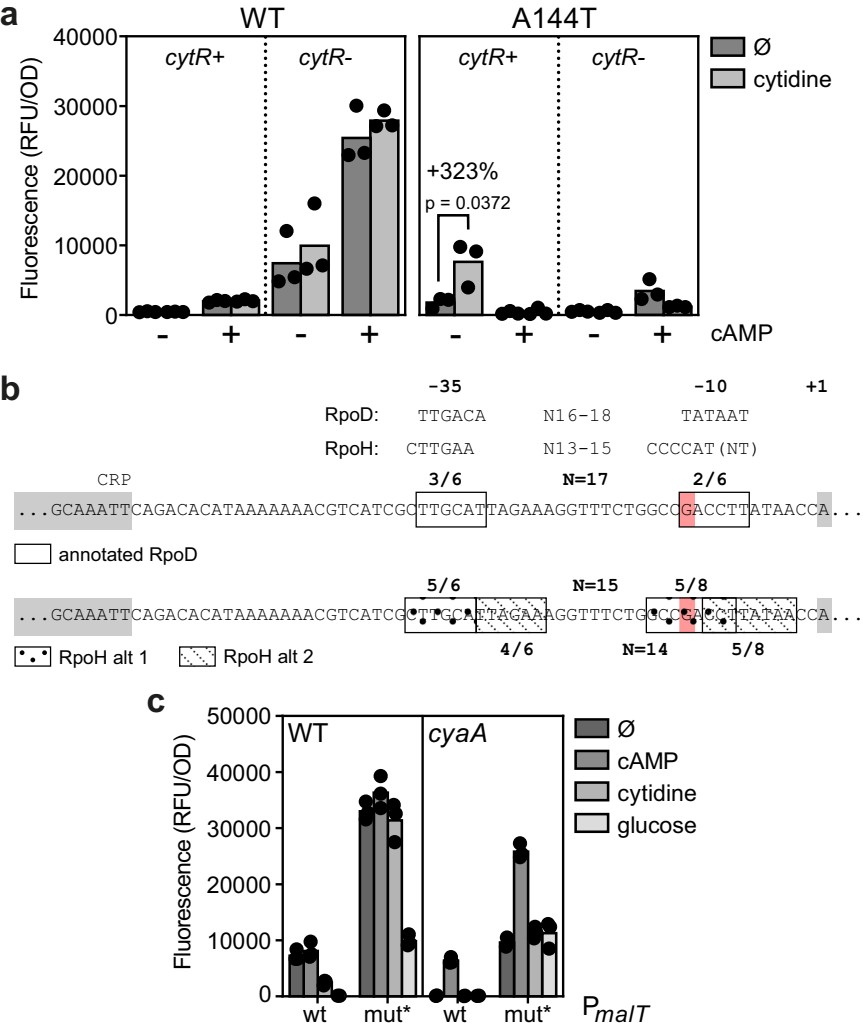

**Fig. 8 The interplay of Crp and co-regulators on the studied promoters. a** Reporter activity from the *rpoH* promoter in MG1655 *cya crp cytR* (+/−) pSEVA-*crp* (Crp WT or A144T) in the presence (light grey) or absence (dark grey, ∅) of cytidine and cAMP (annotated by +/−). Significance was based on two-sided unpaired t-tests between the groups indicated and designated as significant if *p* < 0.05. **b** Annotations of the *malT* promoter with a naturally occurring mutation in red. Top: RpoD and RpoH consensus sequences as reported in litterature[33, 50, 51]. Middle: RpoD annotation as reported in RegulonDB[52]. Bottom: Annotations for hypothetical RpoH recognition sites of the same promoter (two alternatives, alt 1 vs alt 2). **c** Reporter activity in MG1655 (WT or *cya*) from the native *malT* promoter (WT) and from the *malT* promoter with the naturally occurring promoter mutation (mut*, G->T), when supplemented with (dark to light grey) nothing (∅), cAMP, cytidine, or glucose. Data represent the average of three biological replicates. RFU relative fluorescence units, OD optical density.

genes involved in maltose utilization. In the 96 sequenced papillae, one mutation in the promoter region of *malT* was interesting: a G- > T mutation in the −10 box (GACCTT) increases the similarity to the consensus RpoD sequence (TATAAT) (Fig. 8b). When we introduced this mutation into our P*malT-gfp* reporter construct, fluorescence was heavily increased compared to the wildtype promoter construct both in the wild-type and the *cya* background, and the reporter was still responsive to cAMP, but no longer to cytidine (Fig. 8c). This *cis* mutation again points to an important interplay between sigma factors, Crp, and cytidine in maltose utilization.

## Discussion

DNA, RNA, and proteins are the principal informational macromolecules of life, and in all kingdoms of life, signaling pathways have evolved to balance their information flow and synthesis. Energy is harvested from carbohydrates and regulated by a process known as CCR. In *E. coli*, cAMP is a central signaling molecule in this process through its binding to the global transcription factor Crp. Two other purine nucleotides, ppGpp, and pppGpp, play a central role in the stringent response to nutrient availability and like cAMP impact an array of different microbial phenomena such as biofilm formation, persistence, and virulence[30]. The role of pyrimidines as signaling molecules is less well-described. However, pyrimidine biosynthesis flow has been suggested to represent a pivotal sensing mechanism in bacteria, and the build-up of pyrimidines may be a general cellular stress signal[31], but in contrast to the purines little is known about receptors responsible for pyrimidine signal processing. One exception is the transcription factor CytR that binds cytidine and controls the expression of a small set of genes involved in the transport and utilization of nucleosides and deoxynucleosides in *E. coli*[27].

CytR works in concert with Crp[32] and it will be interesting to learn how potential different cytidine binding sites relate to each other in this complex. CytR regulates RpoH and the observations reported here suggest that they both play a bigger role in carbon catabolism than previously known. The sequence determinants in promoters that decide on the recognition by RpoH and RpoD are complex, as the consensus sequences partly overlap while the optimal distances between the −35 and 10 boxes varies[33]. Interestingly, even though the *malT* promoter was not previously reported to be RpoH-regulated, it contains sequences that fit the RpoH consensus better than the RpoD consensus (Fig. 8b), and the *malT* promoter mutation identified and characterized here changes the sequence away from the RpoH towards the RpoD consensus, and renders the corresponding reporter construct cytidine-insensitive. Further studies, such as in vitro transcription of different promoters involved in carbon catabolism using different purified sigma factors in combination with Crp and the core RNA polymerase, could shed further light on a potential broader role of RpoH in carbon metabolism.

The first mechanistic details of gene regulation were revealed by the pioneering work by Jacob and Monod on the *lac* operon[34], one out of hundreds of operons activated by Crp. Yet, while Crp is a paradigm in positive gene regulation, the full details behind CCR are still not fully elucidated. For example, Ullmann, Monod, and co-workers observed that water-soluble extracts of *E. coli* repressed catabolite-sensitive operons such as *lac*[35], and it was later speculated that Crp was negatively regulated by direct binding of a catabolite modulator factor (CMF) to Crp[36]. Studies suggest that α-ketoacids serve a role in inducing CCR[37,38] as one type of CMF, however, a physiologically relevant CMF that directly binds and inhibits Crp has never been identified.

Crp is one of the best studied transcription factors - mutations have been studied for almost five decades and more than 100 different mutants have been identified[15]. cAMP-independent Crp* mutants have also been studied substantially, but our findings provide a clear example that mutants need to be studied in the exact environment they evolve from. The direct effects of uridine and cytidine on Crp are in the mM-range - a relevant physiological concentration at least for uridine[25]. Possibly, cytidine can reach the same concentration level in the spatiotemporal microenvironment of an ageing bacterial colony as evidenced by our metabolomics analysis. Physiological relevance is further supported by the negative effect of *umpH* expression on Crp activity. The frequent isolation of A144T mutations in Crp points to a prominent role for cytidine or uridine as Crp ligands, not just under the specific experimental conditions described here. We find that the canonical A144T mutation converts Crp from being inhibited to being activated by cytidine and uridine. Mutations in *cmk* and the second site mutations that occur sequentially in *crp*A144T and confer independence from cytidine, support the hypothesis that cytidine builds up in the starving, ageing *cya* colonies. Build-up of nucleobases and nucleosides when DNA and RNA are in low demand is previously well-documented e.g. when *E. coli* enters into stationary phase[39], starve[40], and in a phenomenon known as directed overflow metabolism[21]. It is interesting to note that the second site mutation in Q170K is a position very near to an observed second cAMP binding site in Crp[41] and thus it is tempting to speculate that this position may relate to the observed cytidine binding in the A144T mutant.

A simple interpretation of the data presented here is that high concentrations of cytidine impact global transcription as a general stress response by changing the expression of *rpoH* through the CytR transcription factor, thereby acting as the CMF previously searched for by Magasanik, Ullman, Monod, and others[35,36,42]. Cytidine might further impact global gene regulation in mutants such as the *crp*A144T by direct binding to Crp. The regulatory mechanism of α-ketoacids via cAMP[38] and the effects of pyrimidine nucleotides might work in concert in which sensing of anabolic processes are governed by α-ketoacids and sensing of catabolic processes by pyrimidines. This takes Crp to another level as a ubiquitous regulator of transcription in prokaryotes, responding not only to carbon sources, but to the pool of available nucleotides.

In summary, in addition to identifying a surprising role for pyrimidine nucleosides in carbon catabolism, we have demonstrated experimentally that the spatiotemporal dynamics of metabolism can explain transient phenotypes and drive the sequential formation of mutations in microbial evolution. The combination of transcriptional mutagenesis with transiently selectable phenotypes provides a simple mechanism for the formation of multiple sequential mutations that is not in violation with Darwinian evolution theory. Likely this has significance also in the development of cancer in multicellular organisms where the environment is more complex and the lifespan is orders of magnitude greater.

## Methods

**Strains, media and growth conditions**. *E. coli* strains were grown in lysogeny broth (LB) with shaking at 250 rpm or supplemented with agar for growth on plates. All strains are shown in Supplementary Table 3. Strains were incubated at 37 °C unless otherwise mentioned. The antibiotics ampicillin (100 μg/ml), spectinomycin (50 μg/ml), kanamycin (50 μg/ml), or tetracycline (10 μg/ml) were added when needed. MacConkey agar was purchased from BD Diagnostic, Difco MacConkey Agar Base (281810). Simple MacConkey (SM) agar media composition for 300 ml was: Peptone from soybean meat (70178-100 G Sigma Aldrich, 5.1 g), Protease peptone (P0431-250G Sigma Aldrich, 0.9 g), NaCl (Sigma Aldrich, 1.5 g), and Agar (Sigma Aldrich, 4.05 g).

The *crp* gene was deleted in *Escherichia coli* K-12 MG1655 *cyaA::cat* Δ*fnr* and in *Escherichia coli* K-12 MG1655 *cyaA::cat* Δ*fnr cmk*A216E by lambda-red

recombineering with pSIM19 as previously described[43,44] and integration of a *tetA* PCR product with homology ends to the *crp* locus. The *tetA* PCR product was generated with oligo #3734 and #3735 (Supplementary Table 4). With the same method, *cmk, cytR,* and *rpoH* were deleted in K-12 MG1655 *cyaA::cat Δfnr Δcrp*. The *tetA* PCR products were performed with oligos #4010/#4011, #5249/ #5250 and #5290/#5391.

**Plasmid construction.** All plasmid constructs were generated by PCR and USER cloning[45], and verified by sequencing. PCR amplification was performed with either plasmids or genomic DNA as a template. All plasmids are shown in Supplementary Table 3 and oligonucleotides used for plasmid constructions are shown in Supplementary Table 4. The P*trc* promoter in pSEVA27-Crp^ec and pSEVA-Ptrc-*crp*A144T was exchanged with the P*crp* promoter, amplified from genomic DNA with oligonucleotides #4119 and #4120. For construction of pGEM-P_X-hp-sfGFP, the pGEM backbone with *sfgfp-ssrA* was amplified using #4255 and #4257, and the promoters were amplified with #4256/#4258 (P*malT*), #4529/#4530 (P*lacZ*), and #4725/#4726 (P*rpoH*). The naturally occurring variant of pGEM-P*malT*–hp-sfGFP was constructed based on this plasmid using oligos #5601/#5738. The exact promoter sequences applied with the reporter are listed in Supplementary Table 5.

For Crp protein production using pET52b-*crp*, *crp*A144T, the pET52b backbone was amplified with #2697 and #2698 and *crp* was amplified from genomic DNA of wildtype *crp* or *crp*A144T mutant strains using #2710 and #2711. The antibiotic marker was exchanged with an ampicillin resistance marker in pZE21-sfGFP and pZE21-RFP with #4518 and #4519 and backbone amplification with #4516 and #4517.

For fusion of the amplified fragments using USER cloning, backbone and insert were mixed in a 1:3 molar ratio along with 1 μL USER® Enzyme (M5505S, New England Biolabs) and 1 μL T4 ligase buffer (B69, Thermo Scientific), for a total reaction volume of 10 μL. The reaction mixture was incubated for 15 min at 37 °C, 15 min at room temperature, and 10 min at 4 °C, and subsequently transformed into chemically competent DH5α for amplification of plasmids.

**Strain characterization on MacConkey agar.** Strains were first streaked on LB agar supplemented with appropriate antibiotics from −80 °C cryo stocks and the next day restreaked on MacConkey agar supplemented with 1% carbon source and the appropriate antibiotics. Streaking on MacConkey agar was also performed directly from −80 °C cryo. The MacConkey agar plates were further supplemented with either cAMP (0.5 mM), cytidine (10 mM), cytosine (10 mM), cytidine monophosphate (10 mM), or cytidine diphosphate (10 mM).

**Western Blot.** For Western Blot analysis, strains were grown for six hours in 5 mL LB with kanamycin (for *crp* plasmid expression) or without (for native genomic *crp* expression). 2.5 OD units were sampled from each culture and cells were centrifugated at $6000 \times g$ for 10 min. Cell pellets were resuspended in 2x reducing sample buffer to a final concentration of 0.125 OD units/uL and heated to 98 °C for 10 min for protein denaturation. 0.625 OD units were loaded onto a 4–20% Mini-PROTEAN-TGX gel (BioRad, Hercules, CA, USA) and run for 35 min at 174 V. Proteins were transferred to a nitrocellulose membrane using the iBlot dry blotting system (Invitrogen, Thermo Fisher Scientific, Waltham, MA, USA) at 25 V for seven min. The proteins were detected using antigen-specific antibodies. The following antibodies were used: anti-Crp (1:1000; BioLegend, San Diego, CA, USA, Cat#664304, clone#1D8D9), anti-Lep (1:1000; a generous gift from IngMarie Nilsson and Gunnar von Heijne, Stockholm University), anti-mouse (1:25000; Sigma Aldrich, Cat#AP124P, polyclonal) and anti-rabbit (1:50000; Sigma Aldrich). The blot was blocked with 5% w/v skim milk in TBS-T for one hour and incubated overnight with the primary antibody. The blot was washed in TBS-T and incubated for one hour with the secondary antibody. Primary antibodies were diluted in 5% w/v skim milk in TBS-T and secondary antibodies were diluted in TBS-T. The secondary antibody was visualized using Amersham ECL Prime Western Blotting Detection Reagent (GE Healthcare, Chicago, IL, USA). The chemoluminescence signal was detected using G:Box bioimager (Syngene, Cambridge, UK), and the uncropped scans are available in the Source Data file.

**LC-MS targeted metabolite quantification.** Strains were grown for seven hours in 5 mL LB, diluted to $OD_{600}$ 1 in 9 g/L NaCl water and 100 μL of a $10^{-3}$ and $10^{-4}$ dilution was plated on MacConkey medium supplied with or without 1% maltose. Colonies were scraped off agar plates, sampled in ice-cold water after one and five days of incubation at 37 °C and diluted to $OD_{600}$ 7. For sampling of biomass from liquid culture, cultures were inoculated with one colony into a 5 mL SM medium supplied with 1% maltose and sampled after one, three, and six days of incubation at 37 °C. Metabolite extraction was performed in a 96-well Sirocco Protein Precipitation plate (Waters Cooperation, PALL Life Sciences) in 300 μL extraction solvent (methanol-acetonitrile-water 40-40−20% with 0.1% formic acid) on dry ice. Samples with only water or growth medium were added as a control. The samples were filtered by a Positive Pressure-96 (Waters Cooperation) and an additional 200 μL of extraction solvent was filtered. 0.1% ammonium bicarbonate was added to the filtered samples for formic acid neutralization. The extraction solvent was evaporated in Savant SC210A SpeedVac Concentrator (Thermo Scientific) at low

heat overnight and the next day the samples were resuspended in 200 μL Optima® LC-MS water (Fisher Chemical). All strains were grown/plated in two biological replicates and two samples were taken for each time point.

The chromatographic separation was performed using an HPLC (Shimadzu, MD, Columbia) connected to a column of 30 mm, 2.1 mm, 1.8 μm (ACQUITY UPLC HSS T3 Column). Solvent A was 10 mM tributylamine, 2% 2-propanol, 5% methanol, and 10 mM acetic acid and solvent B was 100% 2-propanol. The profile of the gradient followed the specifications as previously described[46] and run time was set at 4.4 min with a constant flow of 0.5 mL/min. Samples were stored in the autosampler at 10 °C, with an injection volume of 10 μL and the column oven was at 40 °C. The detection and quantification of metabolites were achieved by using an AB SCIEX 5500 QTrap (Mass spectrometry, AB SCIEX, Framingham, MA). The instrument was operated in negative mode with a reverse phase ion parring method. The specifications for the instrument were an ionization voltage of −4500, ion source gas 1 and 2 was set to 50 psi, curtain gas of 45 psi, a high collision gas, and a temperature of 500 °C. Each sample contained 13 C labeled internal standard and the intracellular metabolites were absolutely quantified by calculating the ratio between 12 C and 13 C. Prior to the analysis a calibration curve was run containing 102 metabolites. In order to processes the data Smartpeak was used, which is an automated and quantitative data processing application for metabolomics[47].

**Transcriptomics.** Strains were grown in LB broth at 37 °C. After 12 hours of growth, cultures were normalized to $OD_{600}$ using the same growth medium, and $1 \times 10^7$ cells were harvested via centrifugation. Pellets were immediately frozen in liquid nitrogen, and RNA extraction, rRNA removal, and library preparation as well as RNA sequencing (NovaSeq PE150) was subsequently performed by Novogene Inc., Cambridge, UK. All data analysis was performed using CLC Genomics Workbench 20. FastQ files were imported into CLC Genomics Workbench for data processing as paired reads. Reads were trimmed using a quality score limit of 0.05, ambiguous nucleotides (none allowed), and adapters (5′ adapter: AATGATACGGCGACCA CCGAGATCTACACTCTTTCCCTACACGACGCTCTTCCGATCT; 3′ adapter: GATCGGAAGAGCACACGTCTGAACTCCAGTCACATCACGATCTCGTA TGCCGTCTTCTGCTTG). All sequence-based trimming followed standard alignment settings with a mismatch cost of 2, a gap cost of 3, a minimum internal score of 10, and a minimum end score of 4. Trimmed reads were then mapped to a reference sequence (GenBank: U00096.3) using global alignment with standard alignment settings of a match score of 1, a mismatch cost of 2, a linear gap cost of 3, and length fraction and similarity fraction at 0.5 and 0.8, respectively. Following this, the reads were examined for structural variants (quality $p < 0.0001$ and a maximum number of mismatches = 3 for end breakpoints), including deletions and insertions, and based on this information, were realigned using local realignment (two iterations, maximum guidance-variant length 200 bp). Quality distribution followed a PHRED score of average 37 and coverage was on average 99.9% with an average nucleotide distribution of 25% and GC content of ~50%. Read counts for each gene were then converted to TPM values by the software. For subsequent comparison between conditions, samples were normalized based on a statistical model, described in the software manual to consist of TMM normalization[48], calculation of TMM-adjusted log CPM counts, and finally, applied separately for each gene, Z-score normalization (mean = 0, standard deviation = 1). Fold change (FC) values were then calculated and reported as $\log_2(FC)$ for ease of comparison along with *p*-values from the two-sided *t*-test for statistical significance.

All transcriptomics data in this study have been deposited in the European Nucleotide Archive (ENA) and can be accessed via project number: PRJEB46447.

**Protein production.** For *crp* expression, 100 μl chemically competent *E. coli* BL21(DE3) were transformed with 1 μl of pET52b-*crp* or pET52b-*crp*A144T and plated on LB agar. The following day, a colony was inoculated in a preculture of 10 ml LB with 0.4% glucose. The next day, 400 ml LB cultures were inoculated 1:100 with preculture and grown to $OD_{600}$ 0.3-0.5, followed by induction of Crp expression using 1 mM IPTG and 1:800 protease inhibitor. After induction for 4–5 h, the biomass was pelleted at 4 °C ($6000 g$, 10 min), the supernatant discarded, and the pellet saved in stored at −80 °C overnight. Lysis buffer (50 mM $NaH_2PO_4$, 300 mM NaCl, 10 mM imidazole, pH 8) was mixed with lysozyme, benzonase, and protease inhibitor 1:100 immediately before use, and 2 ml was used for resuspension of the cell pellet. The suspension was then put on ice for 2 h, centrifuged at 4 °C ($6000 g$, 30 min), and the supernatant was collected.

Proteins were purified from the supernatant using the Ni-NTA Spin Columns (QIAGEN, Germany) according to the manufacturers protocol, with minor modifications. Equilibration was completed using 5 culture volumes (CV) of lysis buffer, the bound lysate was washed with 10 CV of lysis buffer and 15 CV of buffer NPI-20, and elution was completed with 5.5 CV of buffer NPI-500. Centrifugation was not applied, and flow-through was collected for all steps. An SDS-PAGE was applied to the collected flow through to verify protein production and to determine the concentration of protein, and if necessary, the protein was concentrated using the Amicon Ultra-0.5 Ultracel-10 Centrifugal Filter Units (Merck, Germany) according to the manufacturers protocol.

**Biolayer interferometry**. To study Crp interactions, SA biosensors (ForteBio, USA) were applied using the Octet RED96 system (ForteBio, USA) to assay DNA binding. Prior to use, fresh biosensors were hydrated in 200 μL PBS buffer (from PBS (10X), pH 7.4, Gibco™) for 10 min at 30 °C and 1000 rpm. Assays were run in 96-well plates with one row per sample and protocol steps in separate columns. Biotinylated DNA for the SA biosensors (P*malT*) was generated by annealing oligonucleotides #3915 and #3916 for 20 min at 70 °C. To load the DNA, the SA biosensors were subjected to PBS for a baseline (60 s) before the loading of 20 nM biotinylated DNA (600 s) and quenching using 10 μg/ml biocytin (60 s). The saturated SA biosensors were then subjected to the applied protein buffer PBS-TB (PBS with 0.2% Tween-20 and 0.1% bovine serum albumin) for another baseline (60 s), followed by association using Crp (250 nM) with or without additional analytes (60 s) and dissociation in PBS-TB (60 s). Replicates were made by measuring independent samples of the same reaction.

The biosensor measures the interference of light by molecules immobilized on the biosensor surface. The interference results in a physical displacement of light waves that can be measured in a nanometer, also known as the signal response. The response (nm) parameter is thus the displacement of light during the reaction. For our application, the response rises during the association phase, as proteins bind DNA, thereby causing an increase in the physical displacement of light at the biosensor surface. The final, reported response used for analysis is the average biosensor signal response measured during the final 10 s of the association phase.

**Growth experiments**. Strains were streaked from −80 °C cryo on LB agar with appropriate antibiotics. Next day, three colonies were used for inoculation in 5 ml LB (supplemented with appropriate antibiotics) for biological replicates and grown for 5–7 hours. $OD_{600}$ was adjusted to 1 and 1 μL was used for inoculation in 149 μl medium supplemented with MilliQ water, 0.5 mM cAMP, 10 mM cytidine, 10 mM uridine, or 1% glucose in a 96-well plate (Costar 96 flat bottom) with a breath-seal. The plate was incubated in Synergy H1 Microplate reader from BioTek for 24-48 hours at 37 °C orbital shaking continuously at 425 cpm (3 mm). Optical density (600 nm) and GFP fluorescence (excitation 485 nm, emission 528 nm, gain 50–100, measured from the bottom) was measured every 10 min. Data was outputted in Microsoft Excel 2016 and subjected to pre-processing there, before being transferred to GraphPad Prism 9.1.0 for visualization and statistical analysis.

**Competition assays**. Competition assays were performed with fluorescently labeled strains by growth on agar plates[49]. MG1655 K-12 *cya crp* strains carrying plasmid pSEVA27-*crp*A144T, T140K, or G141D and pZE21-sfGFP or pZE21-RFP were inoculated overnight in 5 ml LB supplemented with ampicillin, kanamycin, and 0.4% glucose. The next day, the cultures were diluted to $OD_{600}$ 2 in LB and plated on SM-maltose agar supplemented with ampicillin and kanamycin, mixed in equal proportions. Plates with only one of the competing strains were also plated. Dilution of cells for plating was done in 9 g/L NaCl water. A 10 μL drop of 0.5 M cytidine was placed in the middle of the plate and plates were incubated overnight at 37 °C. For quantification, cells were scraped off in difference sections of the agar plate, resuspended in 9 g/L NaCl water, adjusted for OD, and fluorescence was measured in the Synergy H1 Microplate reader from BioTek. GFP (excitation 485 nm, emission 528 nm) with gain 100 and RFP (excitation 587 nm, emission 610 nm) fluorescence with gain 90 was measured. The fluorescence values derived from plates with colonies from one strain were used as a reference for quantification (100%).

**Papillation assay**. The MG1655 *cya* strain was grown for seven hours in 5 mL LB at 37 °C, diluted to $OD_{600}$ 1 in sterile 9 g/L NaCl water and 100 μL of a $10^{-4}$ dilution was plated on MacConkey medium supplemented with 1% maltose and the appropriate antibiotics. The plates were placed in plastic boxes containing beakers with water to ensure constant humidity and placed for the duration of the experiment in an incubator at 37 °C. Papillae were restreaked three times before genotype sequencing. The background of the parental strain was verified by sequencing of the *crp* locus before the start of the papillae assay and the phenotypes were tested on MacConkey medium supplied with 1% maltose.

**Resource availability**. Further information and requests for resources and reagents should be directed to and will be fulfilled by the Lead Contact, Morten H.H. Nørholm (morno@biosustain.dtu.dk).

**Reporting summary**. Further information on research design is available in the Nature Research Reporting Summary linked to this article.

## Data availability

Data supporting the findings of this work are available within the paper and its Supplementary Information files. The transcriptomics data generated in this study have been deposited in the European Nucleotide Archive database under project number: PRJEB46447. Source data are provided with this paper.

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

## Acknowledgements
This work was supported by grants from the Novo Nordisk Foundation and the Technical University of Denmark. We would like to thank Andreas Birk Bertelsen for assistance with illustrations, Peter Brodersen and Gunnar von Heijne for comments on the manuscript, and Mogens Kilstrup, Joshua Brickman, and Martin Willemoës for feedback and discussions on the work.

## Author contributions
I.L., P.O.F., S.C., S.W., E.C.F., A.S., A.D., and M.H.H.N. conceptionalized the hypothesis and experimental design. I.L., P.O.F., S.C., S.A.H.H., and, S.B.D. performed the experiments and analysed the data. I.L., P.O.F., A.D., and M.H.H.N wrote the paper with input and corrections from S.W., E.C.F., and A.S. Funding was acquired by M.H.H.N.

## Competing interests
The authors declare no competing interests.
