## [Peer Review File · Nature Communications]

Temporal evolution of master regulator Crp identifies pyrimidines as catabolite modulator factorsREVIEWER COMMENTS

Reviewer #1 (Remarks to the Author):

This work is an interesting extension from authors' previous work (Sekowska et al., 2016) and yields some unexpected discoveries. The mutations were accumulated at *crp* probably due to the strict selection on maltose utilization in the *cya* negative background (cAMP-independent CRP variants will be the main route for cells to survive). By probing differences between wild-type CRP and CRP variants identified, mostly A144T and its derivatives, the changes of CRP properties were characterized. The first half of the work is rather solid indicating that A144T changes the recognition properties of CRP, thus leading to a series of phenotypical differences including pyrimidine (cytidine and uridine) activation. However, the second half of this work about the negative catabolite modulation role of pyrimidine is not fully supported by the reported data. There are some major issues to be addressed:

1. Plasmid-based surrogate assay using *malT* promoter and GFP reporter can be misleading. Most results, if not all, supporting that cytidine and uridine negatively regulate CRP activities are dependent on this surrogate assay. The expression of *malT* is not only under control of CRP but also under control of Mlc and Lrp (how about pyrimidines inhibit these regulators?). In addition, as shown the Fig. S1, uridine didn't significantly inhibit *lacZ* promoter activities, further emphasizing the limitation of the promoter fusion assay using only one responsive promoter. If cytidine and uridine negatively influence CRP activities, then the expression of hundreds of genes controlled by CRP (CRP regulon) will be influenced. More results and relevant information are needed to draw a solid conclusion. Authors cited a paper (reference 25) to support their argument, but it is not sufficient for such an important claim (only ~10 genes from *crp* regulon were mentioned). One simple experiment can be growing wild-type *E. coli* cells on MacConkey plates with different secondary sugars such as lactose, arabinose, xylose, etc., in the presence and absence of cytidine and uridine. If the negative modulation role of cytidine and uridine is valid, the colonies should be consistently unstained. Alternatively, a thorough analysis of expressional changes of CRP regulon (a few hundred genes) in the presence and absence of cytidine and uridine will be informative.
2. If the negative modulation role of pyrimidines is valid, the mechanism is very puzzling. As shown in Fig. 2 and in previous literature, cytidine and uridine are not ligands for CRP. Although authors suggest these pyrimidines may bind CRP and negatively influence its activities, will this conflict with data in Fig. 2? Can this biolayer interferometry provide enough detailed information? Will an electrophoretic mobility shift assay be helpful to test the binding affinity changes between DNA operator sequences and CRP in the presence and absence of cytidine and uridine?

There are also some minor points:

1. Cancer mutational process (mentioned in Abstract and page 2) is quite different from this work here. This mutational hotspot in *crp* is not due to the "stress-induced mutagenesis", but mainly due to the stringent selection. I suggest to delete the association of this work with cancer development.
2. Delete 'nature' on page 4 line 12.
3. Page 7 lines 2, it should be written in past tense.
4. Page 7 line 8, the meaning of "solution space" is not very clear.
5. Page 8 line 1, the phenotypical difference between plasmid-free and plasmid systems was attributed to expression level. However, chromosomal-level expression of *crpA144T* is sufficient to give positive result as shown in Fig 1g (responsive phenotype to *cmk* mutation). The cause can be secondary mutations not the expression level. To back up authors' claim, transcription levels of *crpA144T* need to be measured and compared in two different systems.
6. Fig. 2b, "Response (nm)" (how to calculate) needs to be better explained in Materials and Methods.
7. Page 8, the last sentence, the conclusion "the nucleoside ribose 2'-hydroxyl group is involved in the recognition" could be wrong and other possibilities cannot be excluded. For example, it is possible that methyl group of thymidine has a negative steric effect.
8. Page 9 lines 3 and 5, not sure why authors claim this promoter assay has high sensitivity. The data shown here are not very sensitive (0.25 mM or higher).
9. Page 9 line 3, the statement that "we observed the same effects when replacing *PmalT* with *Plac* in the reporter plasmid" is not correct since uridine has almost no negative effect on *lacZ* promoter

- (Fig. S1b). Authors need to explain the different effect caused by cytidine and uridine.
10. Page 10 line 5, not sure what "in vivo and in vitro difference" precisely means here. Please give more explanation.
 11. The title of Y axis in Fig 4C and S3 is confusing. Please briefly describe the calculation method in figure legend.
 12. Fig. 5, including empty plasmid control in the same strain will provide a comparable baseline especially for Fig. 5b.
 13. Page 13 line 23, please spell out the names for these two CRP homologues.
 14. Page 15 line 6, two other purine nucleotides, which two?
 15. Page 16 line 1, add of between "one" and "the best".
 16. Page 16 line 16, if cytidine and uridine can bind CRP, why data are negative for that in Fig. 2. Please explain this self-conflicting hypothesis.
 17. P17, the first sentence is overstretched from current data (Fig. 5).
 18. Page 18 line 20, what do these number mean? Please cite where these numbers come from.
 19. Page 27 line 11, typo "efer".
 20. How's crp transcript level when provided with cytidine and uridine?

Reviewer #2 (Remarks to the Author):

The authors previously knocked out a gene required for cAMP synthesis in *E. coli* and grew the resulting mutant on a substrate, maltose, that requires cAMP-mediated CRP activation for catabolic pathway expression. In this prior work, the authors identified mutations to *crp* but showed that, when reconstructed in the parental *cya* background, these mutations did not enable growth with maltose. Instead, a second mutation to *cmk* was required. Strains with only a *cmk* mutation spontaneously acquired *crp* mutations.

With all this as background, the authors now sought to understand why the *crp* point mutations were enriched during selection but not sufficient to support growth, and why the *cmk* mutations were beneficial but dependent on a co-occurring *crp* mutation. They showed that the *crp* mutant was activated in the presence of cytidine or CMP, both in vivo and in vitro. This result is consistent with a *cmk* mutation accumulating CMP and activating the mutant Crp. Observed secondary mutations to *crp* were also shown to be constitutively activating. Overall, the authors propose that CMP (or a related metabolite) accumulates in aging colonies. Initial mutations to *crp* allowed activation by this accumulated CMP, but this effect was lost when the cells began to divide and dilute out accumulated CMP. Secondary mutations then enabled growth through constitutive activation of Crp or constitutive accumulation of CMP.

Major comments:

1. I found the narrative disjointed and difficult to follow. Partly, the authors alternated between framing the manuscript around discovery of a new regulatory pathway (involving pyrimidines) versus the characterization of transient, environmentally dependent fitness effects (with vague connections to cancer evolution). Partly, the division between prior work and new results is unclear (with the added challenge that substantial background information from that prior work is required to interpret the new results). And lastly, while I personally am a strong proponent of combining Results and Discussion, this paper formally separates the two but then includes a substantial amount of discussion in the 'Results' section.

I would recommend (1) choosing a single framing and leaving additional topics for a brief section of the discussion; (2) moving all of the background information into the introduction while also editing this section substantially to concisely summarize only the prior work that is necessary to provide context for the new research; and (3) either explicitly combining the Results and Discussion or being more careful about separating them.

2. I always hesitate to suggest new experiments, but it is striking that a paper in which accumulation of pyrimidine metabolites is proposed to play such a critical role never directly measures pyrimidine concentrations. The given explanation is certainly plausible, but it would be much better supported by showing that CMP levels do, in fact, increase in an aging colony and then decrease as the cells are restreaked. Also, Figure S2 suggests that UmpH might affect cytidine/CMP levels as well as UMP/uridine. Differentiating between the two would help to

understand Figure 3c.

3. Similarly, cAMP was varied continuously (e.g. Figure 3A) but, as far as I can tell, never cytidine. Is there a graded response (as might be suggested by the restreaking results of Figure 1D)? How was the value of 10 mM chosen? What is a normal physiological concentration of cytidine, and how much is it changed by the addition of 10 mM extracellular cytidine? Again, I realize that answering these questions might require additional experiments, but so much of the conclusions depend on regulation occurring at physiological pyrimidine concentrations that are never quantified or compared.

4. I find Figure 4C very curious. Why was cytidine spotted at the center of the plate, but then analyzed as +/- cytidine? Presumably a cytidine gradient forms (and then possibly dissipates), and the cells sample a continuous, time-variant concentration. How did the authors choose the boundary of '+-cytidine'? Also, the 'relative fitness' is plotted on the y-axis of Figure 4C, but I believe this is actually 'relative abundance' or such (where 50% abundance implies equal fitness to A144T).

5. The mutations are analyzed genetically, but never discussed in a biochemical context (or connected to decades of research on the biochemistry and functional domain analysis of Crp). Can the authors hypothesize about how/why these mutations might affect the function of Crp?

6. I have concerns about replication and presentation of those replicates. Presenting the mean and standard deviation of two replicates is problematic – you're replacing two data points with two calculated parameters. I would rather just see the raw data. Meanwhile, Figure 2 is based solely on two technical replicates – while I realize the difficulty of performing these experiments, choosing to have no biological replication is concerning. Relatedly, there is no statistical analysis of the results in this manuscript. There's only one reference to significance ("umpH expression significantly reduced expression from the Crp sensitive reporter") and it's not quantified.

Minor comments:

1. Could the results of Figure 1D be due to further mutation accumulation? Was there a difference in growth rate between early and late streaks? Did the authors resequence the later population?

2. Page 7, line 2 – "CrpA144T appears to be more active in a cmkA216E background". More active compared to what? All of Figure 1G is in a cmkA216E background, so is this comparing the results from Figure 1G with another panel?

3. If a figure is divided into panels (e.g. Figure 4), I would prefer that each panel get its own identifier (e.g. panel A is really two figures, a bar chart and the accompanying images. I would describe these as panels A and B).

4. On a practical note, if the authors place figures in-line, I would much prefer that the figure legends accompany them rather than being placed at the end of the manuscript. Line numbers would also be helpful.

Reviewer #3 (Remarks to the Author):

Lauritsen et al. examine the role of hotspot mutations in *E. coli* Crp. In this work, they focus on Crp hotspots and its relation to pyrimidine metabolism. The authors identify Crp mutants and test how those mutants respond in different environments – while modulating pyrimidine levels.

While the novelty of studying Crp mutants is limited, understanding the environmental effect of pyrimidine levels on growth, arrest, and regulator is interesting, and the authors have a unique approach to studying this process using the Crp mutants. Furthermore, the authors do a good job of expanding this study from *ecoli* to another relevant bacterium, *pseudomonas* showing a broad scope. The work is convincing with regards to the role of pyrimidine metabolism in selecting for the phenotype.

However, there are some issues that need to be addressed. Throughout the manuscript, authors raise points regarding the type and source of their mutants. They discuss some basic mechanisms for their mutation, and invoke retromutagenesis as a mechanism for evolving their phenotype without much support. They argue that their mutations are random – the explanation regarding their mutants do not take into account more recent studies.

Overall the work is fairly well-written, but some clarity is needed in their description of the aging experiment and other areas.

Major Comments:

1) The authors state that the mutations occur randomly, but those that enable growth dominate because they are captured by selection. While this may occur, it appears that the mutation types that dominate can be easily explained by mutation pressure alone.

It is well known that C>T mutations are elevated in *E. coli* (Rate and molecular spectrum of spontaneous mutations in the bacterium *Escherichia coli* as determined by whole-genome sequencing - Lee et al., *PNAS*). Furthermore, it is known that 8oxoG in transcribed strands are not random. Oxidative damage is caused at a high rate on the transcribed strand. See transcription associated mutagenesis (TAM) as early as 1997 with many more (Counteraction by MutT Protein of Transcriptional Errors Caused by Oxidative Damage - Taddei et al., *Science*); in yeast (see: Sue Jinks-Robertson).

The mutations they observe are not random and simply explained by general mutation processes whereby G:C>A:T is the most common transition and G:C>T:A is the most common transversion (Lee et al.,). Selection is operating on this phenotype and the likelihood of observing that mutation type is consistent with the site-frequency spectrum of *E. coli*.

To argue for retromutagenesis driving an elevation in these mutation types, the authors would have to show that they are occurring at a rate that exceeds other highly transcribed genes in the genome or other cytosine sites. This can be done by looking at whole genome sequence data at other highly expressed genes/sites. Alternatively, the authors can compare the site-frequency spectrum (SFS) of their mutants against the distribution shown in Lee et al., figure 2 - to argue that there is selection for these mutant types.

More simply, the authors can simply state that the mutations they observe follow the processes described above (Lee et al.). For the "strand bias", A144T/E vs A144K, the authors should focus on the broad literature on transcription associated mutagenesis see above (pg 3/4/13).

2) With regards to secondary mutations, I'd be curious what the SFS of the secondary mutations are. I think this is relevant to claims of retromutagenesis and driver/passenger mutations. This would require the authors to examine the 6 mutation types with conditional to the sites in Crf and compare against Lee et al.

3) The authors phrase a basic question - what makes A144T dominant under these selective conditions? Even the other mutants under "canonical Crp mutations" - show the same pattern and are most commonly characterized Crp mutants (pg 13). One simple explanation is that the number of G>A and C>A mutations generating the desired amino acids are simply the most common.

Oddly enough the authors don't examine T140R which is the most atypical mutant type (G:C>C:G) and observed 16 times. T140R is typed in both column 2 and 4 (not the case for any other locus) which makes me wonder if this mutation was double counted - please explain.

If correct, claims of retromutagenesis might be more supported by this event (16? T140R vs 18 T140K). At least 50% of mutations at this site show phenotype and deviate from SFS shown by Lee et al.

4) Similarly, if correct, the mutation observed at T140K/R forces a change to a positive side charged amino acid. This should be discussed as a unique mutation from A144T/E in "Additional mutations develop sequentially in *crp*" and are likely to be driver genes. Why are Q170K and S62F secondary mutations relevant in that section? - if so cite, or remove.

5) It's unclear how the aging lines were grown - were these independent lines? Please explain the ageing experiment in the methods. Were these all initialized from the same colony? Was the original colony genotyped? You started with three colonies in the "growth experiment", is this the same as the aging lines and how do we know you didn't start with the mutant in your ancestor(s)?

6) Did the authors only sequence those that exhibited papillae phenotype or did they select random aged lines? Did 71 (Table S2) exhibit no mutation but phenotype? This needs to be clearly stated.

7) Are the cmk mutants from the 71 that showed no mutation in crp? This really needs to be clear and an additional column in table S2 that indicate where the cmk mutants are from.

Minor comments:

1) Population heterogeneity and transient phenotypes originating from the same genotypes are increasingly recognized driving forces of evolution, but similar to spatiotemporal microenvironments, the exact conditions in which they evolve are difficult to mimic accurately when studying the causative mutations in isolation.

Please rephrase this paragraph as it is hard to understand and seems out of place. Is this saying that driver mutations are difficult to identify?

2) "Altogether, these observations suggest that the additional crp mutations are not merely passenger mutations, hitchhiking along with Crp* mutations."

Rephrase – are you saying they are hitchhiking or they are not hitchhiking. Can't tell. Also see comment 4 above for a better example of "driver" mutation.

3) "Metabolite in the spatiotemporal microenvironment was building up in aging colonies. Affecting the solution space of Crp mutants."

"and mutation bias caused by the available mutational space could limit the observed solution space."

Solution space – used twice in manuscript. Is this referring to search space or the inhibiting area of metabolite. Maybe just use "inhibiting growth" or? Not sure what the second sentence is referring to, confusing statement in general.

Point-to-point response to reviewers comments

We would like to thank the three reviewers for their very thorough work and positive comments. Several of the suggestions inspired new experimental work and took considerable extra time and effort to resolve, but we feel that this has considerably improved the manuscript and brought us closer to a new understanding of the relationship between Crp and pyrimidines. Our responses to the specific comment are indicated below with an **R and in bold**, and a marked-up version of the revised manuscript has been uploaded together with the manuscript – modifications to the original manuscript are highlighted with track-changes and new sections are marked in yellow. Page and line numbers below refer to the marked-up version.

Reviewers comments:

Reviewer #1 (Remarks to the Author):

This work is an interesting extension from authors' previous work (Sekowska et al., 2016) and yields some unexpected discoveries. The mutations were accumulated at *crp* probably due to the strict selection on maltose utilization in the *cya* negative background (cAMP-independent CRP variants will be the main route for cells to survive). By probing differences between wild-type CRP and CRP variants identified, mostly A144T and its derivatives, the changes of CRP properties were characterized. The first half of the work is rather solid indicating that A144T changes the recognition properties of CRP, thus leading to a series of phenotypical differences including pyrimidine (cytidine and uridine) activation. However, the second half of this work about the negative catabolite modulation role of pyrimidine is not fully supported by the reported data.

There are some major issues to be addressed:

1. Plasmid-based surrogate assay using *malT* promoter and GFP reporter can be misleading. Most results, if not all, supporting that cytidine and uridine negatively regulate CRP activities are dependent on this surrogate assay. The expression of *malT* is not only under control of CRP but also under control of Mlc and Lrp (how about pyrimidines inhibit these regulators?). In addition, as shown the Fig. S1, uridine didn't significantly inhibit *lacZ* promoter activities, further emphasizing the limitation of the promoter fusion assay using only one responsive promoter. If cytidine and uridine negatively influence CRP activities, then the expression of hundreds of genes controlled by CRP (CRP regulon) will be influenced. More results and relevant information are needed to draw a solid conclusion. Authors cited a paper (reference 25) to support their argument, but it is not sufficient for such an important claim (only ~10 genes from *crp* regulon were mentioned). One simple experiment can be growing wild-type *E. coli* cells on 1 MacConkey plates with different secondary sugars such as lactose, arabinose, xylose, etc., in the presence and absence of cytidine and uridine. If the negative modulation role of cytidine and uridine is valid, the colonies should be consistently unstained. Alternatively, a thorough analysis of expressional changes of CRP regulon (a few hundred genes) in the presence and absence of cytidine and uridine will be informative.

R Thanks a lot for these positive and inspiring comments. In retrospect, we fully agree that the presented data did not fully support the important claim of pyrimidines playing a role in carbon metabolism. As suggested, we screened for pyrimidine effects on

MacConkey agar supplemented with a number of different carbon sources. This data is now included in a new Figure 3 and Suppl. Fig 3, and provide additional support for the global effect of cytidine as a signal with substantial impact on carbon metabolism. These findings also inspired us to explore further the potential link between the CytR transcription factor and CCR and in the revised manuscript we present new data suggesting that both CytR and the heat shock sigma factor RpoH play a role in coupling cytidine sensing with Crp-signaling. It is true that the negative uridine-effect is not significant in the PlacZ data presented in the previous Supplementary figure 1 (now 4). In the many different assays we have employed to demonstrate the effect of nucleosides on genes involved in carbon metabolism, we often see stronger effects of cytidine than uridine. This is likely due to the partial involvement CytR that only binds cytidine.

2. If the negative modulation role of pyrimidines is valid, the mechanism is very puzzling. As shown in Fig. 2 and in previous literature, cytidine and uridine are not ligands for CRP. Although authors suggest these pyrimidines may bind CRP and negatively influence its activities, will this conflict with data in Fig. 2? Can this biolayer interferometry provide enough detailed information? Will an electrophoretic mobility shift assay be helpful to test the binding affinity changes between DNA operator sequences and CRP in the presence and absence of cytidine and uridine?

R Again a valid point and we agree that the in vitro data presented (and thus support for direct binding) cannot stand on its own. Possibly, the in vitro activation of the CrpA144T mutant reflects a general increase in sensitivity towards nucleotides and may not be physiologically relevant. Crp mutants have previously been shown to be activated by other nucleotides such as adenosine and cGMP (this is now added to the introduction). With many additional experiments we now instead provide further in vivo evidence that pyrimidines affect carbon metabolism broadly and that CytR and RpoH are involved.

There are also some minor points:

1. Cancer mutational process (mentioned in Abstract and page 2) is quite different from this work here. This mutational hotspot in crp is not due to the “stress-induced mutagenesis”, but mainly due to the stringent selection. I suggest to delete the association of this work with cancer development.

R This point was also brought up by reviewer #2 (see below). We do think that it is important (and well-documented) to discuss the link between cancer and adaptive mutations in microbes, but have now changed the narrative to more accurately reflect the new findings in the study and removed the association to cancer in the abstract.

2. Delete ‘nature’ on page 4 line 12.

R The sentence was modified.

3. Page 7 lines 2, it should be written in past tense.

R The sentence was modified.

4. Page 7 line 8, the meaning of “solution space” is not very clear.

R We clarified the concept by extending to “evolutionary solution space”

5. Page 8 line 1, the phenotypical difference between plasmid-free and plasmid systems was attributed to expression level. However, chromosomal-level expression of *crpA144T* is sufficient to give positive result as shown in Fig 1g (responsive phenotype to *cmk* mutation). The cause can be secondary mutations not the expression level. To back up authors' claim, transcription levels of *crpA144T* need to be measured and compared in two different systems.

R Fair point. We have now measured transcription levels by RNAseq and protein levels by western blotting, which confirmed the higher expression of *crp* from the plasmid. The data is included in a new supplementary Fig 1 and the data is discussed in the main text on lines 197-198.

6. Fig. 2b, "Response (nm)" (how to calculate) needs to be better explained in Materials and Methods.

R This response is not calculated but a physical parameter measured by the biosensor. We have added a more thorough explanation in the Materials and Methods section.

7. Page 8, the last sentence, the conclusion "the nucleoside ribose 2'-hydroxyl group is involved in the recognition" could be wrong and other possibilities cannot be excluded. For example, it is possible that methyl group of thymidine has a negative steric effect.

R We agree with this point and removed the claim

8. Page 9 lines 3 and 5, not sure why authors claim this promoter assay has high sensitivity. The data shown here are not very sensitive (0.25 mM or higher).

R Thank you for pointing this out. The reporter has a high dynamic range as it expresses at very low levels in the absence of cAMP and highly at high concentrations of cAMP. We modified the description of the reporter accordingly.

9. Page 9 line 3, the statement that "we observed the same effects when replacing PmalT with Plac in the reporter plasmid" is not correct since uridine has almost no negative effect on *lacZ* promoter (Fig. S1b). Authors need to explain the different effect caused by cytidine and uridine.

R We have now clarified the statement in the main text as referring to the cytidine effect and not uridine. It is correctly observed that uridine does not have a significant effect on expression from the *lacZ* promoter. However, we cannot provide a sufficient explanation for this. We have consistently observed the uridine effect as being equal to or lesser than the cytidine effect across multiple promoters (see also above for further discussion on this point). As we hypothesize about the interaction between Crp and nucleosides, the supposed effect of such an interaction may very well depend on the individual promoter structures and the location of the Crp binding site in relation to the transcription start site, which varies considerably within the Crp regulon genes. Obviously, the indicated involvement of CytR may also explain the stronger effect of cytidine in some cases. In the interest of transparency, we did not want to remove the *PlacZ*+uridine data from the manuscript, as others may be able to elucidate on this effect.

10. Page 10 line 5, not sure what “in vivo and in vitro difference” precisely means here. Please give more explanation.

R The sentence was modified (line 353) and the additional work e.g. on the connection with CytR should also make this point clearer now.

11. The title of Y axis in Fig 4C and S3 is confusing. Please briefly describe the calculation method in figure legend.

R We added a description of the approach and the calculation to the legend

12. Fig. 5, including empty plasmid control in the same strain will provide a comparable baseline especially for Fig. 5b.

R This data seems less relevant now that links to CytR and RpoH have been established. (The inhibition may simply be due to the same interactions between the Pseudomonas homologs and e.g. RpoH). We have therefore chosen to remove this.

13. Page 13 line 23, please spell out the names for these two CRP homologues.

R Description removed as explained above.

14. Page 15 line 6, two other purine nucleotides, which two?

R This may be a slightly confusing (but nevertheless standard) description – we spelled out the two nucleotides.

15. Page 16 line 1, add of between “one” and “the best”.

R Done

16. Page 16 line 16, if cytidine and uridine can bind CRP, why data are negative for that in Fig. 2. Please explain this self-conflicting hypothesis.

R We modified this claim to better fit the data

17. P17, the first sentence is overstretched from current data (Fig. 5).

R The sentence was removed

18. Page 18 line 20, what do these number mean? Please cite where these numbers come from.

R A reference to the supplementary table describing these numbered oligonucleotides is present just above these numbers

19. Page 27 line 11, typo “after”.

R Thanks. The typo was corrected

20. How’s *crp* transcript level when provided with cytidine and uridine?

R This is an interesting remark. It is entirely possible that *crp* itself is affected by pyrimidines as it is well-documented that CRP regulates its own synthesis. We did an attempt to explore this by performing RNAseq in the presence and absence of cytidine, but observed no significant changes in *crp* expression. We added the data as

supplementary Figure 6 and discuss the data in the main text on lines 351-352

Reviewer #2 (Remarks to the Author):

The authors previously knocked out a gene required for cAMP synthesis in *E. coli* and grew the resulting mutant on a substrate, maltose, that requires cAMP-mediated CRP activation for catabolic pathway expression. In this prior work, the authors identified mutations to *crp* but showed that, when reconstructed in the parental *cya* background, these mutations did not enable growth with maltose. Instead, a second mutation to *cmk* was required. Strains with only a *cmk* mutation spontaneously acquired *crp* mutations.

With all this as background, the authors now sought to understand why the *crp* point mutations were enriched during selection but not sufficient to support growth, and why the *cmk* mutations were beneficial but dependent on a co-occurring *crp* mutation. They showed that the *crp* mutant was activated in the presence of cytidine or CMP, both in vivo and in vitro. This result is consistent with a *cmk* mutation accumulating CMP and activating the mutant Crp. Observed secondary mutations to *crp* were also shown to be constitutively activating. Overall, the authors propose that CMP (or a related metabolite) accumulates in aging colonies. Initial mutations to *crp* allowed activation by this accumulated CMP, but this effect was lost when the cells began to divide and dilute out accumulated CMP. Secondary mutations then enabled growth through constitutive activation of Crp or constitutive accumulation of CMP.

Major comments:

1. I found the narrative disjointed and difficult to follow. Partly, the authors alternated between framing the manuscript around discovery of a new regulatory pathway (involving pyrimidines) versus the characterization of transient, environmentally dependent fitness effects (with vague connections to cancer evolution). Partly, the division between prior work and new results is unclear (with the added challenge that substantial background information from that prior work is required to interpret the new results). And lastly, while I personally am a strong proponent of combining Results and Discussion, this paper formally separates the two but then includes a substantial amount of discussion in the 'Results' section.

I would recommend (1) choosing a single framing and leaving additional topics for a brief section of the discussion; (2) moving all of the background information into the introduction while also editing this section substantially to concisely summarize only the prior work that is necessary to provide context for the new research; and (3) either explicitly combining the Results and Discussion or being more careful about separating them.

R Thank for suggesting this, which relates somewhat to minor comment (1) brought up by reviewer #1. We have removed the connection to cancer in the abstract and we have removed one of the paragraphs on retromutagenesis from the introduction. This way we believe the manuscript focuses more on the surprising link between pyrimidines and carbon metabolism.

Although we agree that the start of the results section is unconventional due to the close relation to prior work, we also provide new phenotypic characterizations of the mutants isolated previously. This makes it difficult to move the text to the introduction (it would also be strange to have new results in the introduction). For example, the dominant A144T mutation was described previously but the transient phenotype was only realized later. Another example is the double mutations in Crp – this was described previously, but the connected accelerated evolution of single mutants

such as A144T was not. The third example is the mutations in *cmk* – again these were previously observed but here we studied them further by reintroducing *crp* on plasmids for phenotypic characterization. We believe it serves a purpose to describe these new observations the same place as details of the previous work is introduced. Besides, this is only the case for the first couple of paragraphs in the results section.

2. I always hesitate to suggest new experiments, but it is striking that a paper in which accumulation of pyrimidine metabolites is proposed to play such a critical role never directly measures pyrimidine concentrations. The given explanation is certainly plausible, but it would be much better supported by showing that CMP levels do, in fact, increase in an aging colony and then decrease as the cells are restreaked. Also, Figure S2 suggests that UmpH might affect cytidine/CMP levels as well as UMP/uridine. Differentiating between the two would help to understand Figure 3c.

R Thank you for this excellent suggestion. As we also comment to Reviewer #1 major point (1), we fully agree that the presented data did not fully support the important claim of pyrimidines playing a role in carbon metabolism. We addressed the specific comment on pyrimidines levels brought up here by metabolomics sampling at different time points both from agar plates and liquid cultures. These experiments confirm that the relevant pyrimidines increase in abundance over time. These data are now included as a new Figure 2, Supplementary Figure 2, and are described in the main text on pages X. lines 201-259

3. Similarly, cAMP was varied continuously (e.g. Figure 3A) but, as far as I can tell, never cytidine. Is there a graded response (as might be suggested by the restreaking results of Figure 1D)? How was the value of 10 mM chosen? What is a normal physiological concentration of cytidine, and how much is it changed by the addition of 10 mM extracellular cytidine? Again, I realize that answering these questions might require additional experiments, but so much of the conclusions depend on regulation occurring at physiological pyrimidine concentrations that are never quantified or compared.

R Indeed, we also varied cytidine levels. We have now included the data as supplementary figure 5 and discuss the data on lines 349-351 and measured nucleotide levels by metabolomics as discussed above.

4. I find Figure 4C very curious. Why was cytidine spotted at the center of the plate, but then analyzed as +/- cytidine? Presumably a cytidine gradient forms (and then possibly dissipates), and the cells sample a continuous, time-variant concentration. How did the authors choose the boundary of '+-cytidine'? Also, the 'relative fitness' is plotted on the y-axis of Figure 4C, but I believe this is actually 'relative abundance' or such (where 50% abundance implies equal fitness to A144T).

R We modified the description and labeling to better explain the approach, which is copied from a previously published protocol.

5. The mutations are analyzed genetically, but never discussed in a biochemical context (or connected to decades of research on the biochemistry and functional domain analysis of Crp).

Can the authors hypothesize about how/why these mutations might affect the function of Crp?

R Thanks this is a great suggestion. We added paragraphs discussing the biochemical context of the Crp mutations to the introduction and the discussion - lines 97-100 and 619-622

6. I have concerns about replication and presentation of those replicates. Presenting the mean and standard deviation of two replicates is problematic – you’re replacing two data points with two calculated parameters. I would rather just see the raw data. Meanwhile, Figure 2 is based solely on two technical replicates – while I realize the difficulty of performing these experiments, choosing to have no biological replication is concerning. Relatedly, there is no statistical analysis of the results in this manuscript. There’s only one reference to significance (“umpH expression significantly reduced expression from the Crp sensitive reporter”) and it’s not quantified.

R We agree and modified the plots to include the raw data and included the statistics.

Minor comments:

1. Could the results of Figure 1D be due to further mutation accumulation? Was there a difference in growth rate between early and late streaks? Did the authors resequence the later population?

R All mutants were sequenced after several rounds of restreaking. This was exactly how we discovered the phenomenon that the A144T phenotype was transient. We do observe additional mutations occurring (such as Q170K described in the manuscript), but these are more active CRP variants which indicates that the environment primarily selects for increased rather than decreased maltose fermentation.

2. Page 7, line 2 – “CrpA144T appears to be more active in a cmkA216E background”. More active compared to what? All of Figure 1G is in a cmkA216E background, so is this comparing the results from Figure 1G with another panel?

R Thank for pointing this out. CrpA144T, both provided on plasmid and encoded in the genome, is white on maltose MacConkey. This can be seen in Figure 1H. We have now pointed this out in the text on 164-166

3. If a figure is divided into panels (e.g. Figure 4), I would prefer that each panel get its own identifier (e.g. panel A is really two figures, a bar chart and the accompanying images. I would describe these as panels A and B).

R This is a good suggestion. We added the panel indications.

4. On a practical note, if the authors place figures in-line, I would much prefer that the figure legends accompany them rather than being placed at the end of the manuscript. Line numbers would also be helpful.

R In principle, we agree with both points. We added the line numbers. However, with the number of figures and the level of detail in the figure legends, the main text becomes dispersed over many pages and we feel this disturbs the flow of reading. Thus, we have kept the legends at the end of the manuscript.

Reviewer #3 (Remarks to the Author):

Lauritsen et al. examine the role of hotspot mutations in *E. coli* Crp. In this work, they focus on Crp hotspots and its relation to pyrimidine metabolism. The authors identify Crp mutants and test how those mutants respond in different environments – while modulating pyrimidine levels.

While the novelty of studying Crp mutants is limited, understanding the environmental effect of pyrimidine levels on growth, arrest, and regulator is interesting, and the authors have a unique approach to studying this process using the Crp mutants. Furthermore, the authors do a good job of expanding this study from *ecoli* to another relevant bacterium, *pseudomonas* showing a broad scope. The work is convincing with regards to the role of pyrimidine metabolism in selecting for the phenotype.

However, there are some issues that need to be addressed. Throughout the manuscript, authors raise points regarding the type and source of their mutants. They discuss some basic mechanisms for their mutation, and invoke retromutagenesis as a mechanism for evolving their phenotype without much support. They argue that their mutations are random – the explanation regarding their mutants do not take into account more recent studies.

Overall the work is fairly well-written, but some clarity is needed in their description of the aging experiment and other areas.

Major Comments:

1) The authors state that the mutations occur randomly, but those that enable growth dominate because they are captured by selection. While this may occur, it appears that the mutation types that dominate can be easily explained by mutation pressure alone.

It is well known that C>T mutations are elevated in *ecoli* (Rate and molecular spectrum of spontaneous mutations in the bacterium *Escherichia coli* as determined by whole-genome sequencing - Lee et al., *pnas*). Furthermore, it is known that 8oxoG in transcribed strands are not random. Oxidative damage is caused at a high rate on the transcribed strand. See transcription associated mutagenesis (TAM) as early as 1997 with many more (Counteraction by MutT Protein of Transcriptional Errors Caused by Oxidative Damage – taddei et al., *science*); in yeast (see: Sue Jinks-Robertson).

The mutations they observe are not random and simply explained by general mutation processes whereby G:C>A:T is the most common transition and G:C>T:A is the most common transversion (lee et al.,). Selection is operating on this phenotype and the likelihood of observing that mutation type is consistent with the site-frequency spectrum of *ecoli*.

To argue for retromutagenesis driving an elevation in these mutation types, the authors would have to show that they are occurring at a rate that exceeds other highly transcribed genes in

the genome or other cytosine sites. This can be done by looking at whole genome sequence data at other highly expressed genes/sites. Alternatively, the authors can compare the site-frequency spectrum (SFS) of their mutants against the distribution shown in Lee et al., figure 2 – to argue that there is selection for these mutant types.

More simply, the authors can simply state that the mutations they observe follow the processes described above (lee et al.). For the “strand bias”, A144T/E vs A144K, the authors should focus on the broad literature on transcription associated mutagenesis see above (pg 3/4/13).

R Thank you for these highly interesting comments. We did not intend to argue that retromutagenesis drives specific nucleotide mutations – only the bias in detecting the mutations on the transcriptional strand, but we acknowledge that the way it was phrased could be interpreted like that. We removed this part of the introduction to focus more on the novel findings presented in the manuscript on the connection between pyrimidines and CCR. See below for further comments.

2) With regards to secondary mutations, I'd be curious what the SFS of the secondary mutations are. I think this is relevant to claims of retromutagenesis and driver/passenger mutations. This would require the authors to examine the 6 mutation types with conditional to the sites in Crf and compare against lee at al.

R Again we agree that this would be highly interesting. However, in response to the comments from the other reviewers we have chosen to focus the manuscript more towards the surprising general role of pyrimidines in carbon metabolism. In order to do so, we found it necessary to downplay the description of the underlying molecular mechanisms in evolution such as retromutagenesis.

3) The authors phrase a basic question – what makes A144T dominant under these selective conditions? Even the other mutants under “canonical Crp mutations” – show the same pattern and are most commonly characterized Crp mutants (pg 13). One simple explanation is that the number of G>A and C>A mutations generating the desired amino acids are simply the most common.

R Thank you for pointing this out. We rephrased the question to “What makes A144T dominant over other Crp* mutations...”as some of these are generated by the same nucleotide mutations.

Oddly enough the authors don't examine T140R which is the most atypical mutant type (G:C>C:G) and observed 16 times. T140R is typed in both column 2 and 4 (not the case for any other locus) which makes me wonder if this mutation was double counted – please explain.

R Thank you for pointing this out. Indeed this was a mistake in the table that we have now corrected. We found the other mutations more relevant to compare with A144T because they were generated by the same nucleotide mutations and because G141D is the most frequently isolated mutation together with A144T (in the literature).

If correct, claims of retromutagenesis might be more supported by this event (16? T140R vs 18 T140K). At least 50% of mutations at this site show phenotype and deviate from SFS shown by lee et. al.

R This is a very interesting suggestion. We are actively working in the lab to provide further evidence on the importance of retromutagenesis, but have decided to focus entirely on pyrimidines and carbon metabolism in this manuscript.

4) Similarly, if correct, the mutation observed at T140K/R forces a change to a positive side charged amino acid. This should be discussed as a unique mutation from A144T/E in “Additional mutations develop sequentially in *crp*” and are likely to be driver genes. Why are Q170K and S62F secondary mutations relevant in that section? – if so cite, or remove.

R They are relevant because one of the main driving questions behind our work was why these mutations develop after the A144T mutation – and we picked Q170K because it was the most frequent mutation co-occurring with A144T.

5) Its unclear how the aging lines were grown – were these independent lines? Please explain the ageing experiment in the methods. Were these all initialized from the same colony? Was the original colony genotyped? You started with three colonies in the “growth experiment”, is this the same as the aging lines and how do we know you didn’t start with the mutant in your ancestor(s)?

R Yes, they were independent lines and we always make sure the mutations have not occurred prior to these long-term experiments (by phenotyping on MacConkey and sequencing). We added the info to the materials and methods.

6) Did the authors only sequence those that exhibited papillae phenotype or did they select random aged lines? Did 71 (Table S2) exhibit no mutation but phenotype? This needs to be clearly stated.

R Yes all exhibited papillation phenotype and some had no *crp* mutation. We have added the description to the table legend.

7) Are the *cmk* mutants from the 71 that showed no mutation in *crp*? This really needs to be clear and an additional column in table S2 that indicate where the *cmk* mutants are from.

R No, but this is a good point. *cmk* mutations were found in 24 of the 96 sequenced genomes and always co-occurred with the *crp* mutations A144T, A144E, or T140R. We have added the information in the main text. All mutations in the 96 sequenced genomes are available in reference 13.

Minor comments:

1) Population heterogeneity and transient phenotypes originating from the same genotypes are increasingly recognized driving forces of evolution, but similar to spatiotemporal microenvironments, the exact conditions in which they evolve are difficult to mimic accurately when studying the causative mutations in isolation.

Please rephrase this paragraph as it is hard to understand and seems out of place. Is this saying that driver mutations are difficult to identify?

R We agree and removed the paragraph.

2) “Altogether, these observations suggest that the additional *crp* mutations are not merely passenger mutations, hitchhiking along with *Crp** mutations.”

Rephrase – are you saying they are hitchhiking or they are not hitchhiking. Can’t tell. Also see comment 4 above for a better example of “driver” mutation.

R Thank you for pointing this out. We rephrased the sentence to: “Altogether, these observations suggest that the additional *crp* mutations are not merely hitchhiking along with *Crp mutations”**

3) “Metabolite in the spatiotemporal microenvironment was building up in aging colonies. Affecting the solution space of *Crp* mutants.”

“and mutation bias caused by the available mutational space could limit the observed solution space.”

Solution space – used twice in manuscript. Is this referring to search space or the inhibiting area of metabolite. Maybe just use “inhibiting growth” or? Not sure what the second sentence is referring to, confusing statement in general.

R Thank you for pointing this out. We added “search” space to clarify.

REVIEWERS' COMMENTS

Reviewer #1 (Remarks to the Author):

The points previously proposed by me have been addressed positively. There are a lot of new data which strengthen the manuscript, but meanwhile they complicate the research narrative too. Especially for the newly added RpoH portion. I feel the merits are sufficient for this current version and further revisions may not be necessary considering the timeline and efforts made from the authors.

Reviewer #2 (Remarks to the Author):

In general, the authors have addressed my concerns. The addition of the metabolite data really pins down a key element of the story. My only remaining issue is with the introduction, which I still struggle with. I don't think that it's absolutely necessary to revise, but I personally would encourage reframing it.

Major comments:

1) I still find the introduction out of place with the rest of the paper. The title and abstract focus on pyrimidines as regulatory factors, but then the introduction starts talking about cancer. Then after a switch to discussing Crp, the introduction closes rather abruptly with 'we found some things that warrant investigation'. It's not wrong, per se, but it's not really helping to prepare me for the rest of the paper. I'd really suggest rewriting the introduction to better set up the meat of the paper, rather than this ancillary story about cancer evolution.

Minor comments:

- 1) Line 84 – 'seeked' should probably be 'sought'.
- 2) Line 160 – 'two-three fold' is confusing. Perhaps 'two- to three-fold'?
- 3) Figure 2: This is just personal preference, but I have trouble, at a glance, telling which of the bars in 2b correspond to a given x-axis label. It would help me if you made the pairs of bars close within a strain versus between strains.

Reviewer #3 (Remarks to the Author):

Lauritsen et al. examine the role of hotspot mutations in *E. coli* global transcription factor Crp, examining the sequence of mutations that arise in Crp. They previously explore *ecoli* colonies in *cya* background and examine a cAMP independent route to maltose formation, usually A144T (and G141D). They further explore what the sequence of mutations are that lead to A144T.

They find that genes involved in pyrimidine metabolism are mutational hotspots and that *cmk* and second-site mutations occur with *crp* mutations. They find that A144T mutation converts Crp from being inhibited to being activated. Second they find that mutations in *cmk* and second site mutations in pyrimidine linked genes.

They argue that the build up of pyrimidines (from aging) alters the evolution of the genes. The work supports its conclusions and claims using a number of microbiological tests, and the authors fixed a number of flaws in the tables and clarify their methods in the latest revision. The significance of this work is that they show strong selection for these genes suggesting and that this can be an evolutionary driver as bacteria age (due to build-up of these pyrimidine products).

The authors do a good job of addressing reviewer comments including performing an additional experiment to show pyrimidine metabolism, clarifying where the mutations are coming from, and removing some speculative discussion on the mechanisms driving these hotspots.

There is enough detail for the work to be reproduced.

Line 424 – issues with spacing.

POINT_BY_POINT RESPONSE TO REVIEWERS' COMMENTS

Responses to the specific comments are highlighted with Rs and in blue font below. Changes in the manuscript are highlighted in yellow and track-changes in the accompanying marked-up manuscript.

Reviewer #1 (Remarks to the Author):

The points previously proposed by me have been addressed positively. There are a lot of new data which strengthen the manuscript, but meanwhile they complicate the research narrative too. Especially for the newly added RpoH portion. I feel the merits are sufficient for this current version and further revisions may not be necessary considering the timeline and efforts made from the authors.

Reviewer #2 (Remarks to the Author):

In general, the authors have addressed my concerns. The addition of the metabolite data really pins down a key element of the story. My only remaining issue is with the introduction, which I still struggle with. I don't think that it's absolutely necessary to revise, but I personally would encourage reframing it.

Major comments:

1) I still find the introduction out of place with the rest of the paper. The title and abstract focus on pyrimidines as regulatory factors, but then the introduction starts talking about cancer. Then after a switch to discussing Crp, the introduction closes rather abruptly with 'we found some things that warrant investigation'. It's not wrong, per se, but it's not really helping to prepare me for the rest of the paper. I'd really suggest rewriting the introduction to better set up the meat of the paper, rather than this ancillary story about cancer evolution.

R: We modified the introduction so that it now follows the following logical flow:

(1) A (shortened) paragraph on why it is of general interest to study hotspot mutations that occur in structured environments such as ageing bacterial colonies (further downplaying the link to cancer as suggested).

(2) This is followed by a detailed description of the role of Crp in CCR and the hotspot mutations previously identified in *crp*.

(3) We then introduce the mutations previously found in pyrimidine metabolism – this section is moved from the results section and this modification therefore also serves to better separate previous results from the current work – as suggested by the editor and previously by one of the reviewers.

(4) In response to the suggestions here, we now also included a short new introduction to pyrimidine metabolism to make the introduction better aligned with the title and abstract as suggested by the reviewer here. To this end, we also changed the order of appearance of subpanels in Fig. 1 and supplementary figures (7 became 1).

(5) Finally, in line with the instruction from the editor, we have modified the final sentence so that it briefly describes the new work.

Minor comments:

1) Line 84 – 'seeked' should probably be 'sought'.

R: We have modified the entire paragraph as described above

2) Line 160 – ‘two-three fold’ is confusing. Perhaps ‘two- to three-fold’?

R: We modified as suggested

3) Figure 2: This is just personal preference, but I have trouble, at a glance, telling which of the bars in 2b correspond to a given x-axis label. It would help me if you made the pairs of bars close within a strain versus between strains.

R: We modified as suggested

Reviewer #3 (Remarks to the Author):

Lauritsen et al. examine the role of hotspot mutations in *E. coli* global transcription factor Crp, examining the sequence of mutations that arise in Crp. They previously explore *ecoli* colonies in *cya* background and examine a cAMP independent route to maltose formation, usually A144T (and G141D). They further explore what the sequence of mutations are that lead to A144T.

They find that genes involved in pyrimidine metabolism are mutational hotspots and that *cmk* and second-site mutations occur with *crp* mutations. They find that A144T mutation converts Crp from being inhibited to being activated. Second they find that mutations in *cmk* and second site mutations in pyrimidine linked genes.

They argue that the build up of pyrimidines (from aging) alters the evolution of the genes. The work supports its conclusions and claims using a number of microbiological tests, and the authors fixed a number of flaws in the tables and clarify their methods in the latest revision. The significance of this work is that they show strong selection for these genes suggesting and that this can be an evolutionary driver as bacteria age (due to build-up of these pyrimidine products).

The authors do a good job of addressing reviewer comments including performing an additional experiment to show pyrimidine metabolism, clarifying where the mutations are coming from, and removing some speculative discussion on the mechanisms driving these hotspots.

There is enough detail for the work to be reproduced.

Line 424 – issues with spacing.

R: This is a reference not formatted correctly as pointed out by the editor as well. We corrected the formatting